# Loss of the abasic site sensor HMCES is synthetic lethal with the activity of the APOBEC3A cytosine deaminase in cancer cells

Josep Biayna[1], Isabel Garcia-Cao[2], Miguel M. Álvarez[1], Marina Salvadores[1], Jose Espinosa-Carrasco[1], Marcel McCullough[1], Fran Supek[1,3]*, Travis H. Stracker[2,4]*

**1** Genome Data Science, Institute for Research in Biomedicine (IRB Barcelona), The Barcelona Institute of Science and Technology, Barcelona, Spain, **2** Genomic Instability and Cancer, Institute for Research in Biomedicine (IRB Barcelona), The Barcelona Institute of Science and Technology, Barcelona, Spain, **3** Catalan Institution for Research and Advanced Studies (ICREA), Barcelona, Spain, **4** National Cancer Institute, Center for Cancer Research, Radiation Oncology Branch, Bethesda, Maryland, United States of America

* fran.supek@irbbarcelona.org (FS); travis.stracker@nih.gov (THS)

**Data Availability Statement:** All relevant data are within the paper and its Supporting Information files. Analysis in Fig 3 used data from screens in

## Abstract

Analysis of cancer mutagenic signatures provides information about the origin of mutations and can inform the use of clinical therapies, including immunotherapy. In particular, APO-BEC3A (A3A) has emerged as a major driver of mutagenesis in cancer cells, and its expression results in DNA damage and susceptibility to treatment with inhibitors of the ATR and CHK1 checkpoint kinases. Here, we report the implementation of CRISPR/Cas-9 genetic screening to identify susceptibilities of multiple A3A-expressing lung adenocarcinoma (LUAD) cell lines. We identify HMCES, a protein recently linked to the protection of abasic sites, as a central protein for the tolerance of A3A expression. HMCES depletion results in synthetic lethality with A3A expression preferentially in a TP53-mutant background. Analysis of previous screening data reveals a strong association between A3A mutational signatures and sensitivity to HMCES loss and indicates that HMCES is specialized in protecting against a narrow spectrum of DNA damaging agents in addition to A3A. We experimentally show that both HMCES disruption and A3A expression increase susceptibility of cancer cells to ionizing radiation (IR), oxidative stress, and ATR inhibition, strategies that are often applied in tumor therapies. Overall, our results suggest that HMCES is an attractive target for selective treatment of A3A-expressing tumors.

## Introduction

The apolipoprotein B mRNA-editing enzyme catalytic polypeptide-like 3 (APOBEC3) family of cytidine deaminases is a major source of mutagenesis in human cancers. Elevated mRNA levels of APOBEC3A (A3A) and APOBEC3B (A3B) enzymes, as well as an activating germline polymorphism in the *A3A* and *A3B* genes, were associated with a particular mutational

Olivieri et al. that is available in Mendeley data and a link is provided in the manuscript.

**Funding:** J.B., M.M. and F.S. were supported by the ERC Starting Grant (757700 "HYPER-INSIGHT", to F.S). I.G.C. was funded by an AECC fellowship. M.M.A. and J.E.C. were supported by the Spanish Ministry of Science, Innovation and Universities (BFU2017-89833-P "RegioMut", to F.S.). T.H.S. was funded by the Spanish Ministry of Science, Innovation and Universities (MCIU: PGC2018-095616-B-I00/GINDATA and FEDER) and the Intramural Research Program of the National Institutes of Health, National Cancer Institute. The T.H.S. and F.S. labs are supported by the Centres of Excellence Severo Ochoa award and the CERCA Programme. F.S. is funded by the ICREA Research Professor programme. The funders had no role in study design, data collection and analysis, decision to publish, or preparation of the manuscript.

**Competing interests:** The authors have declared that no competing interests exist.

**Abbreviations:** A3A, APOBEC3A; B, APOBEC3B; APOBEC3, apolipoprotein B mRNA-editing enzyme catalytic polypeptide-like 3; ATCC, American Type Culture Collection; BER, base excision repair; BIR, break-induced replication; DOX, doxycycline; DPC, DNA–protein crosslink; ds, double-strand; DSB, double-strand break; gDNA, guide DNA; gRNA, guide RNA; HA, haemagglutinin; HR, homologous recombination; ICL, interstrand crosslink; IR, ionizing radiation; LFC, $\log_2$ fold change; LUAD, lung adenocarcinoma; MGMT, O-6-methylguanine-DNA methyltransferase; MLE, maximum-likelihood estimation; MMR, mismatch repair; MOI, multiplicity of infection; NER, nucleotide excision repair; NGS, next-generation sequencing; NHEJ, non-homologous end joining; NSCLC, non-small cell lung cancer; PC, principal component; PIKK, PI-3 kinase-like kinase; PPP2R2A, Protein phosphatase 2A subunit; QC, quality control; qRT-PCR, quantitative real-time PCR; RRA, robust rank aggregation; SD, standard deviation; SDSA, synthesis-dependent strand annealing; sgRNA, single gRNA; TLS, translesion synthesis; TMZ, temozolomide; TOP1, Topoisomerase I.

signature of C-to-T and C-to-G changes in a TCW trinucleotide context (where W is A or T) [1–4]. Both A3A and A3B have been implicated in localized hypermutation, which can occur in 2 different patterns: the focused kataegis ("mutation showers," likely occurring during repair of DNA double-strand (ds) breaks (DSBs) [1,5]) and the diffuse omikli pattern ("mutation fog," proposed to occur during repair of mismatched or damaged nucleotides [6,7]). The A3s are a cause of intratumor genetic heterogeneity and generate driver mutations in tumors [7–10]. Consistently, A3 mutagenesis has prognostic value in cancers [1,11–13]. Recent genomics work suggests that A3 mutagenesis appears rare in various types of apparently noncancerous somatic cells [14]; moreover, A3 mutagenesis appears to increase in intensity in metastatic cancers [15]. This suggests that vulnerabilities of APOBEC-expressing cells would provide a window of opportunity to selectively target certain types of tumor cells while sparing their healthy counterparts.

Overexpressing A3 enzymes in yeast and human cell lines result in clustered mutation patterns, resembling those seen in cancer genomes [13,16,17]. Therefore, such experimental models of A3 overexpression appear useful for recapitulating DNA damaging and mutagenic effects that occur in tumors due to APOBEC activity. The A3A mutagenesis signature is distinguishable from that of A3B, and both signatures are present in varying proportions across cancer types. However, the A3A signature is predominant overall [7,18–21] consistent with experiments suggesting that A3A induces high levels of DNA damage [2,22]. We therefore focused our attention on A3A.

A3s deaminate cytosine in DNA to generate uracil, which can be converted to an abasic (AP) site, following the action of uracil glycosylases [23]. Uracil is mutagenic, causing U:G mispairing during copying. Moreover, AP sites cannot be directly copied by the replicative DNA polymerases during S-phase, necessitating the use of potentially mutagenic translesion synthesis (TLS) polymerases [24]. A3A-induced damage occurs during S-phase, and AP sites can lead to replication fork stalling and replication stress [25–27]. Processing of AP sites by AP endonucleases can allow repair by the base excision repair (BER) pathway. This can promote further A3 mutagenesis, particularly if coupled with the activity of DNA mismatch repair (MMR) that can "hijack" BER intermediates [6]. Alternatively, the processing of AP sites in ssDNA can convert them to DNA DSBs, a more cytotoxic lesion. Processing and repair of DSBs by the homologous recombination (HR) or break-induced replication (BIR) pathways generates additional ssDNA which may be targeted by APOBECs [5,28]. Thus, multiple DNA repair pathways are engaged as a consequence of A3-induced DNA damage, and activity of these pathways can promote further A3 DNA damage.

Increased reliance of some tumors on particular DNA repair pathways has long been exploited as a therapeutic avenue. For example, brain cancers that lose activity of the O-6-methylguanine-DNA methyltransferase (MGMT) enzyme, which can directly reverse O-6 adducts, are more sensitive to the DNA-methylating drug temozolomide (TMZ) [29]. Ovarian and breast tumors with failures in HR repair pathways due to inactivated BRCA1 and BRCA2 genes are more sensitive to PARP inhibitors, such as Olaparib [30,31]. These examples of successful therapeutic applications encouraged us to search for targetable DNA repair pathways in cancer cells exposed to increased A3A activity.

Overexpression of A3A causes DNA damage and replication stress; the latter can be targeted by inhibitors of the ATR and CHK1 checkpoint kinases that respond to replication stress [22,32,33]. The observation that cell cycle checkpoint inhibitors enhanced the levels of DNA damage in A3A-expressing cells indicates that it is plausible that they have many additional inherent vulnerabilities that can be therapeutically exploited, apart from the replication stress response, which is a more general phenomenon not specific to A3A.

We performed a CRISPR/Cas-9-based genome-wide screen for genes required to tolerate A3A-mediated DNA damage in a panel of cell lines from non-small cell lung cancer (NSCLC), where APOBEC activity has been shown to play an important role in tumor evolution [3,4,34–36]. Among other hits, we identified factors involved in multiple DSB repair pathways, including RAD9A, a component of the 9-1-1 alternative clamp loader and the recently characterized MCM8-MCM9-HROB complex [37–40]. Crucially, we found that different genetic backgrounds are consistently and strongly dependent on the gene encoding HMCES (5hmC binding, embryonic stem cell–specific protein) for cell viability [8,15] under APOBEC stress, but not otherwise. Recently, HMCES (also known as SRAP Domain-containing Protein 1), as well as the related bacterial protein YedK, were shown to covalently bind to AP sites in ssDNA, where they act as a suicide enzyme to protect them from TLS or AP endonucleases [41–46]. In addition, HMCES has been proposed to function in the repair of DSBs in the canonical and the alternative non-homologous end joining (NHEJ) pathways [42,47,48]. We validated HMCES depletion as a sensitizer to A3A in multiple cell lines, consistent with recent work [49], and show that HMCES limits DNA damage and prevents loss of cell viability resulting from *A3A* expression in a manner specific to TP53-mutant cells. Together, our results identify additional druggable targets to be considered in A3A-expressing cancer cells and establish a central role for HMCES in preventing the toxicity of A3A expression.

## Results

### Generation of non-small cell lung cancer cell lines with inducible A3A expression

To examine the influence of A3A expression in NSCLC, we established a panel of cell lines with doxycycline (DOX)-inducible expression of a haemagglutinin (HA)-tagged A3A using the pSLIK-Neo vector system (Fig 1A and 1B) [27]. This included NCI-H358, LXF-289, A549, and a TP53 null variant of A549, A549$^{TP53−/−}$, generated using CRISPR/Cas-9 targeting (S1 Fig). Treatment of cells with DOX resulted in a dose-dependent increase in A3A mRNA expression and protein levels (Fig 1A and 1B). Consistent with previous reports that A3A expression caused DNA damage and cell cycle checkpoint activation, we observed a slower growth rate in several of the NSCLC cell lines (Fig 1C) [27].

### CRISPR/Cas-9 genetic screen for A3A synthetic lethality

In order to identify vulnerabilities of A3A-expressing cells, we performed genome-wide CRISPR/Cas-9 screening in 3 lung adenocarcinoma (LUAD) cell lines: LXF-289, A549, and A549$^{TP53−/−}$. Screening was performed at an established IC$_{25}$ dose of DOX for A549 and A549$^{TP53−/−}$ and IC$_{25}$ and IC$_{50}$ doses for LXF-289 (Fig 2A). The cell lines, with or without pSLIK-Neo A3A, were transduced with a single-vector lentiviral library expressing Cas-9 (Brunello) and guide RNAs (gRNAs) for 19,114 genes (4 single gRNAs (sgRNAs) per gene) at a multiplicity of infection (MOI) ≤0.4 (Fig 2A) [50]. Following puromycin selection, cells were lysed, genomic DNA extracted, and preparation and analysis of initial gRNA representation (T0) was performed. Cells were subsequently expanded for 15 days and treated with DOX to induce A3A expression. At multiple time points following DOX treatment, we collected cells and amplified guide DNAs (gDNAs) using barcoded primers. DNA was sequenced and analyzed for changes in abundance of gDNAs targeting various genes, comparing the DOX-treated cells with the untreated (control) cell line at the same time point using the MAGeCK-RRA tool [51], thus revealing genes which have stronger fitness effects in A3A-expressing cells. Additionally, we compared to T0 to determine overall essential genes. The

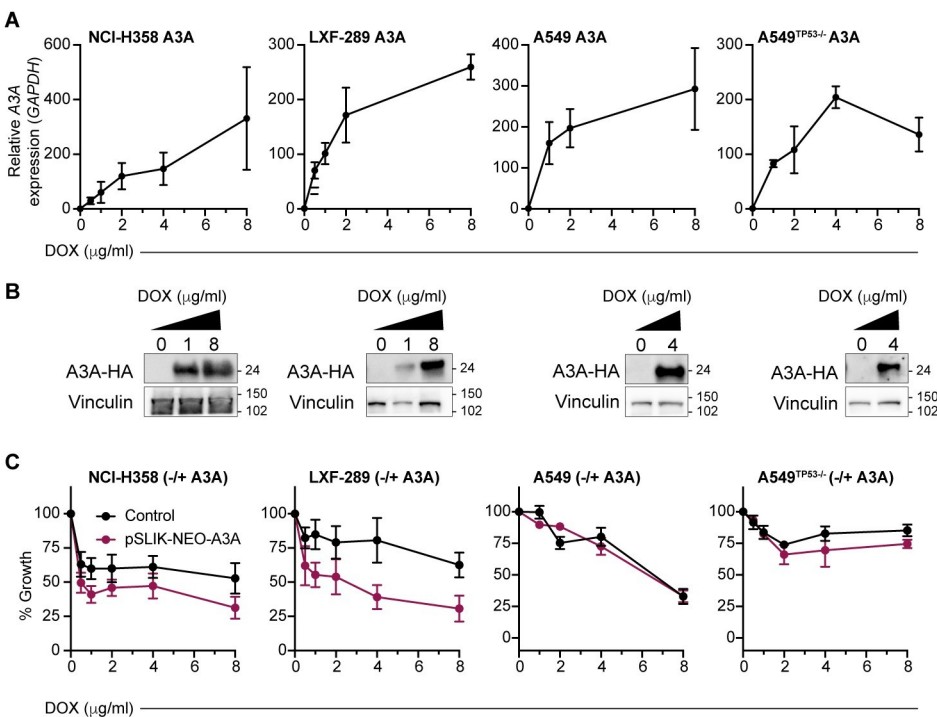

**Fig 1. Inducible A3A expression reduces the fitness of LUAD cell lines. (A)** A3A mRNA levels in NCI-H358, LXF-289, A549, and A549$^{TP53-/-}$ cell lines transduced with a DOX-inducible A3A cassette at various concentrations of DOX. qRT-PCR and western blot analysis were repeated 2 times. (**B**) Western blot detection of HA-A3A upon DOX induction. LXF-289, NCI-H358, A549, and A549$^{TP53-/-}$ cells were collected and lysed 72 h posttreatment. Vinculin serves as a loading control, and molecular mass is indicated in kilodaltons. (**C**) Growth (percentage of growth rate relative to the cells without DOX) for the indicated cell lines after 72 h of DOX treatment measured with AlamarBlue. In red, cells transduced with the inducible A3A cassette, and in black, the parental cell line (no A3A) exposed to the same concentration of DOX. Growth assays were repeated 2 (A549$^{TP53-/-}$) or 3 times (all other lines). For all graphs, mean and SEM are shown. Uncropped western blots are provided in S1 Raw Images, and numerical data underlying plots are provided in S1 Data. A3A, APOBEC3A; DOX, doxycycline; HA, haemagglutinin; LUAD, lung adenocarcinoma; qRT-PCR, quantitative real-time PCR.

control experiment showed that DOX itself (in a genetic background lacking the A3A plasmid) affected the essentiality of very few genes (S2 Fig).

As TP53 status has been shown to influence CRISPR/Cas-9 screening results, we first compared A549$^{TP53-/-}$ with the LXF-289 cell line that bears a TP53 mutation (c.742C>T; p. R248W per DepMap.org record ACH-000787) (Fig 2B) [54]. We prioritized genes by an overall APOBEC essentiality score: average log$_2$ fold change (LFC) over 6 measurements: 3 time points for the A549$^{TP53-/-}$ cell line (T9, T12, and T15) and 3 for the LXF-289 cell line (T5, T10, and T15). The top 5 hits by this score were the genes coding for the AP site protecting protein HMCES [42], the RAD9A cell cycle checkpoint control protein, the MCM8 component of the MCM8-MCM9-HROB complex[37,38,55], ATXN7L3, a component of the SAGA chromatin modifying complex[56,57], and HGC6.3, an uncharacterized protein. For four of the 5 genes, the individual gRNAs, 4 per gene, consistently sensitized cells to A3A expression (Fig 2B, S3 Fig). However, HGC6.3 guides displayed an inconsistent temporal trend, and the effect size for HGC6.3 was very different across the 2 cell lines (S3 Fig). An additional analysis by the MAGeCK-MLE method [51] (S4 Fig) suggested that all 4 gRNAs for *HGC6.3* had low knock-out efficiency (all < = 0.72; S1 Table) in contrast to other top hits, and we thus disregarded HGC6.3 in further analysis. The remaining 4 top hits did not show clear differences in effects

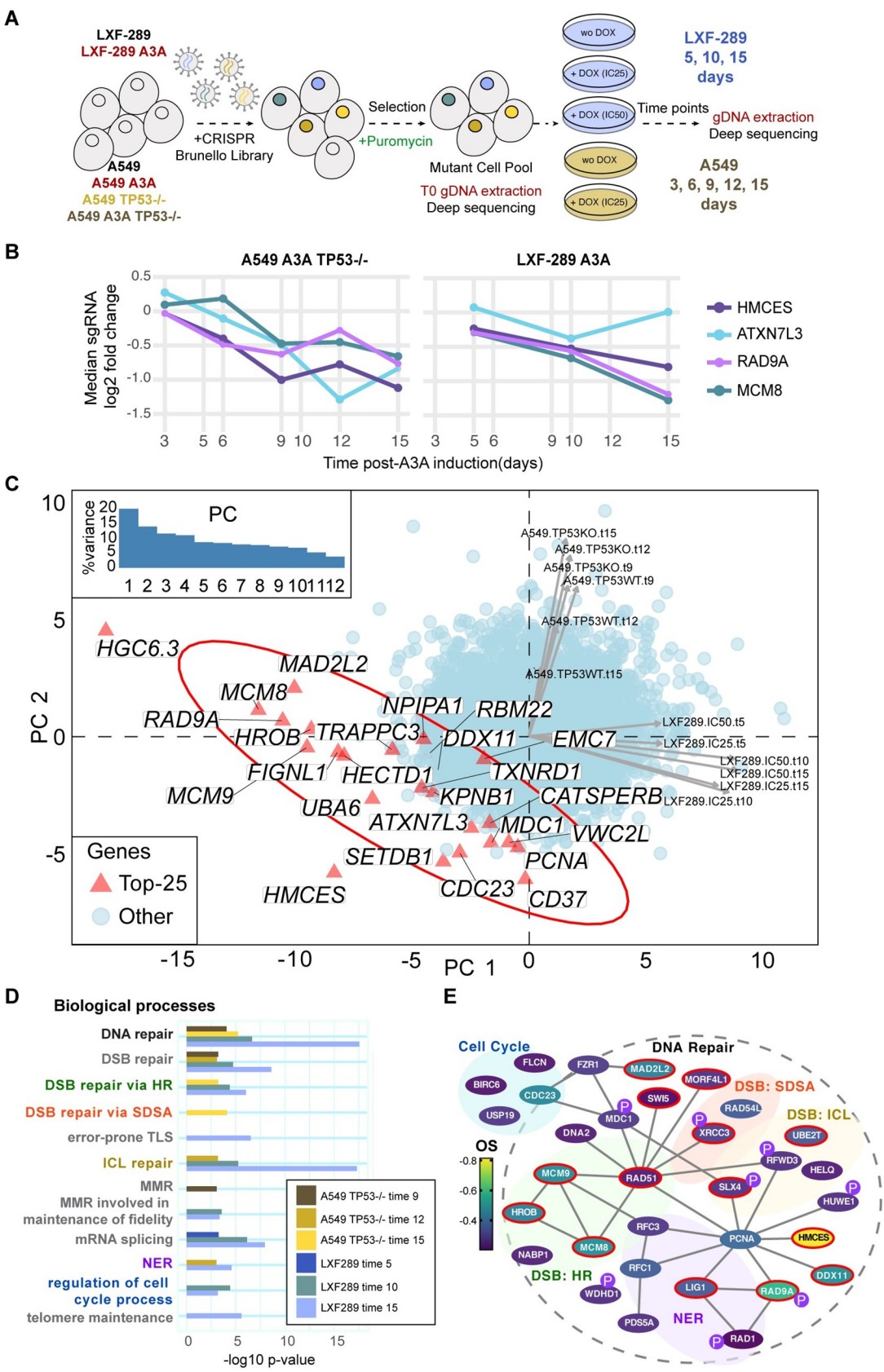

**Fig 2. CRISPR/Cas-9 genetic screen indicates HMCES and other DNA repair genes as vulnerabilities of A3A-expressing cells.** (**A**) Experimental design using the Brunello genome-wide library [50]. (**B**) Depletion of the sgRNAs targeting top 4 genes upon A3A overexpression, as prioritized by the overall A3A conditional essentiality score: LFC across 3 time points of the LXF-289 cell line and the 3 latest time points of the A549$^{TP53-/-}$ cell line. The y-axis shows the median of the 4 sgRNAs per gene. (**C**) PC analysis of A3A conditional LFC scores for all genes across all 12 experimental conditions (see labels next to arrows, which show loadings of the conditions on PC1 and PC2; "KO" implies TP53−/− and "WT" TP53 wild-type A549 cell line; numbers in labels are the time points; "IC25" and "IC50" are 2 concentrations of DOX in the LXF-289 cell line). The top 25 genes, prioritized by the same A3A conditional score as in panel **B**, are highlighted on the figure. Inlay shows a scree plot, with the amount of variance explained by the 12 PCs. The LXF-289-specific *HGC6.3* hit is likely an artefact (S1 Table; see Results text). LFC for all genes/sgRNAs are included in S2 Table and S2 Data. LFC for sgRNAs (from the top 100 genes plus nontargeting), shown in S2A–S2C Fig, are included in S3 Data. (**D**) Gene Ontology enrichment analysis of the top hits in the 6 experiments considered for the overall A3A conditional score (as in panel **B**). Plot shows −log10 *p*-value (unadjusted) from the GORILLA server. Underlying data and full Gene Ontology analysis and category names are provided in S3 Table. (**E**) Network schematic of cell cycle and DNA repair–related genes from the top 300 hits in the screen (OS). Genes identified in the Gene Ontology analysis and appearing in the top 300 genes by OS are shown. Color denotes the OS, lines indicate physical interactions (thebiogrid.org) [52], and a red border indicates they were identified in the Gene Ontology analysis in both cell lines. Those gene products identified as PIKK (ATM, ATR, and DNA-PKcs) targets are indicated with a purple "P," data from PhosphoSitePlus [53]. Numerical data for all graphs in the figure are provided in S3 Table and S2 Data. A3A, APOBEC3A; DSB, double-strand break; HR, homologous recombination; ICL, interstrand crosslink; LFC, log$_2$ fold change; MMR, mismatch repair; NER, nucleotide excision repair; OS, overall score; PC, principal component; PIKK, PI-3 kinase-like kinase; SDSA, synthesis-dependent strand annealing; sgRNA, single gRNA; TLS, translesion synthesis.

between the 2 tested A3A dosages in the LXF-289 cell line (corresponding to IC$_{25}$ and IC$_{50}$) (panel D in S3 Fig). Next highest-ranking hits included the UBA6 ubiquitin-activating enzyme and a further 5 genes that were all related to DNA repair, DNA replication, or cell cycle control (*DDX11*, *MCM9*, *CDC23*, *MAD2L2* (also known as *REV7*), and *HROB* (also known as *C17ORF53* or *MCM8IP*); S1 Table).

## Distinct genes but consistent pathways in A3A responses across genetic backgrounds

We further examined global trends in response to A3A expression across all approximately 19,000 genes and 12 different experimental conditions, using principal component (PC) analysis (Fig 2C). This suggested that globally, the results differ considerably between the A549 and LXF-289 cell lines, indicating that the genetic background modulates conditional essentiality of many genes under A3A conditions (data for top hits shown in S5 Fig). In particular, the first 2 PCs explained 34% variability in the data and separated the LXF-289 from the A549 cell line data points, but they did not appreciably separate (i) the 3 different time points within each cell line, nor (ii) the TP53 wild-type versus TP53$^{-/-}$ background of the A549 cell line, nor (iii) the 2 different A3A doses (IC$_{25}$ and IC$_{50}$) in the LXF-289 cell line. The same PC analysis highlighted 2 genes with an extremely strong signal in the A3A response: *HMCES*, because it is consistently observed across both genetic backgrounds (Fig 2C), and the LXF-298-specific *HGC6.3* gene, which we suspect is an artefact (see above). We further substantiated these results using MAGeCK-MLE [51]; in this analysis, *HMCES* was the only gene which was conditionally essential (>2 standard deviations (SDs) away from the mean of the beta coefficients, per MAGeCK-MLE recommendation) in late time point samples in both A549$^{TP53-/-}$ and LXF-289 cells (S4 Fig).

Despite the apparent differences in the effects of individual genes between A549$^{TP53-/-}$ and LXF-289 cells (S2 Table), Gene Ontology analysis yielded consistent results (Fig 2D), identifying DNA repair–related pathways as strongly enriched. In both cell lines, DSB repair was a major enriched biological process, with HR representing the predominant pathway, and to a lesser extent, interstrand crosslink (ICL) repair (at $p < 10^{-3}$ using the GORILLA server; see S3 Table for list of results). Furthermore, nucleotide excision repair (NER) and DNA MMR were enriched in both cell lines, as well as the regulation of the cell cycle (Fig 2D, S3 Table). In LXF-

289 cells, the NHEJ and Fanconi anemia pathways were strongly represented among the top hits enriched, as well as *MCM8*, *MCM9*, and *HROB*, genes that have previously been implicated in HR (Fig 2D, S3 Table) [37,38,55]. Further enriched pathways related to DNA repair included error-prone TLS and telomere maintenance in LXF-289 cells. Intriguingly, there was also a strong enrichment of mRNA splicing genes (S3 Table). In A549$^{TP53-/-}$ cells, there was also an enrichment of DSB repair via synthesis-dependent strand annealing (SDSA) (Fig 2D, S3 Table). Overall, we conclude that A3A expression induces dependencies on a variety of DNA repair and related pathways in cells, some of which may be specific to certain genetic backgrounds, while others appear more universal.

## Analyses of large-scale genetic screening data suggest a unique role of HMCES

Our genetic screens performed in different cancer cell lines yielded many A3A conditionally essential genes that were specific to one of the 2 cell lines (Fig 2C). Quality control (QC) parameters of the screening data indicated the high quality of all the screens by gDNA representation, by the ability to discriminate common essential genes and by the separation between APOBEC conditionally essential genes and nontargeting control sgRNAs (S6 Fig, S4 Table). Therefore, a likely explanation for the differences between cell lines could be that the genetic background and/or epigenetic state of a cell line determines its complement of essential genes upon A3A activation.

This motivated us to seek further evidence that the top hits we observed across both cell lines would indeed be valid across a wider spectrum of genetic backgrounds. To this end, we analyzed data from 76 LUAD, lung squamous cell carcinoma, and head and neck squamous cell carcinoma cell lines (thus approximately matching our experimental models by tissue or cell type) from the Project Achilles database [58,59]. In particular, we searched among the top 10 genes from our experiments for correlations between the burden of A3 context mutations in the cell line exomes and the essentiality of a gene. By this metric, the *HMCES* gene obtained the highest scores in the external Project Achilles data (Fig 3A, S5 Table) for APOBEC signature 13 and signature 2 (slope of fit −0.29 and −0.5, respectively; combined $p = 0.03$, $t$ test on the regression coefficient, 1-tailed). In contrast, the *MCM8*, *RAD9A*, and *ATXN7L3* genes, even though observed in both cell lines in our experiments, did not score highly in this analysis (Fig 3A; we note that *MCM8* does rank more highly than other top hits from the genetic screen, but is nonetheless not robustly supported; S5 Table). This provided additional confidence that the synthetic interaction between *HMCES* and A3 activity is likely to hold across very diverse genetic backgrounds, as it is observed across a large cell line panel. A caveat of this analysis is that the A3 mutational signature may reflect past activity or intermittent activity of A3, and thus the lack of correlation in this analysis does not necessarily rule out the validity of the hit.

In addition to HMCES, the analysis of our genetic screening data revealed many common hits participating in DSB repair (Fig 2E). While it is likely that AP sites resulting downstream of APOBEC ssDNA lesions may generate DSBs in need of repair, such hits in the screen would plausibly also result from other agents inducing DSBs. Because our screening effort is focused on finding potentially actionable vulnerabilities, we were less interested in finding hits that result from DNA damaging conditions in general, which may abundantly occur also in healthy cells and are not linked to a genetic marker, in contrast to APOBEC activity, which may be more common in tumors and is evident in mutational signatures. We therefore analyzed data from previous genetic screens performed in the RPE1$^{TP53-/-}$ cell line under a variety of different genotoxic agents [60]. We found that some of the common hits for A3A are also sensitizers

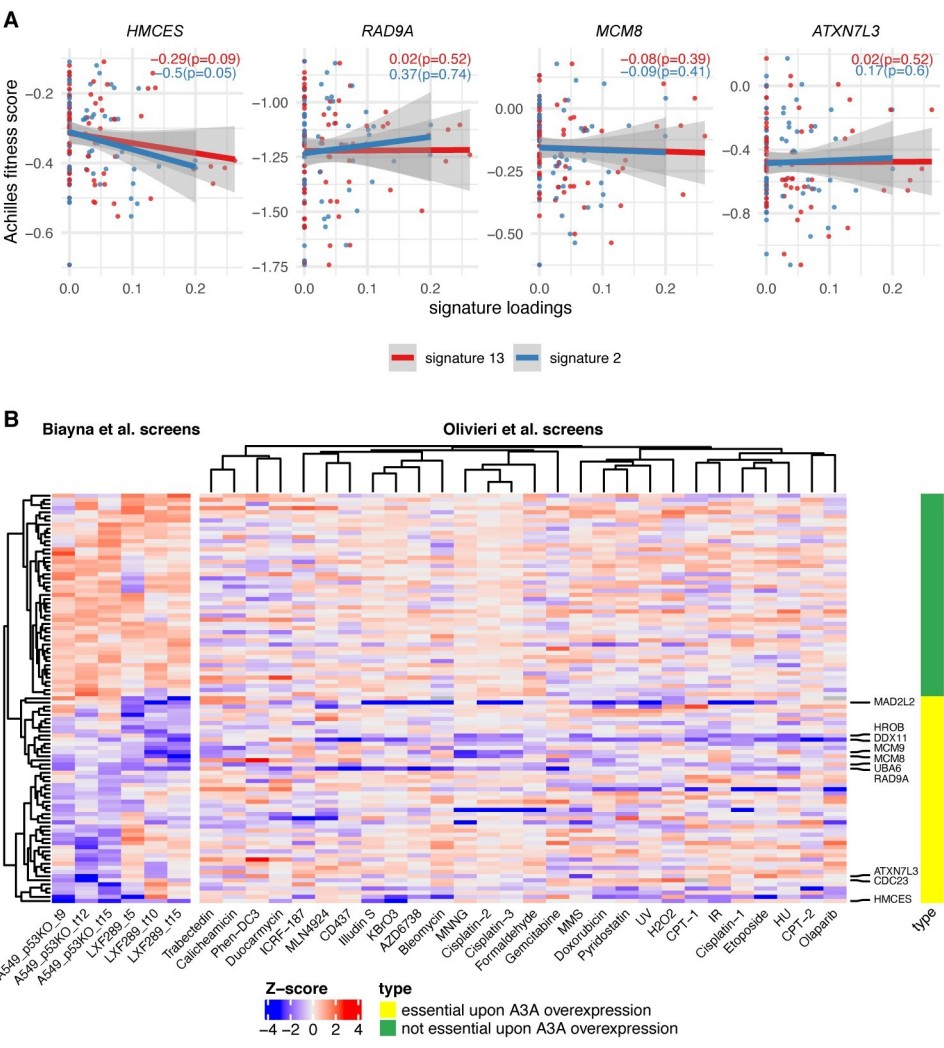

**Fig 3. Dependency on HMCES is associated with mutational signatures of APOBEC across 76 lung and head and neck cancer cell lines.** (**A**) Gene essentiality fitness score from Project Achilles vs. APOBEC mutational signatures exposures, for cell lines from head and neck squamous cell carcinoma, LUAD, and lung squamous cell carcinoma, in four of the genes with the greatest overall score in our screens; see S5 Table and S7 Fig for associations with additional prominent genes. The slope and *p*-value (1-tailed, lower) for the regression model for both A3 signatures are shown within each panel. The more negative the slope, the more sensitive the cell lines are to the depletion of the gene at a higher level of the APOBEC signature. (**B**) Heatmap shows a gene-level normalized LFC (gene essentiality score) upon A3A overexpression for 2 cell lines and for 3 time points (Biayna et al. screens); right panel shows Z-scores of gene essentiality after genotoxin exposure (Olivieri et al. screens[60]). Data for 50 genes that are essential upon A3A overexpression in our screens (i.e., genes with the most negative mean LFC across 6 data points) and 50 nonessential genes upon A3A overexpression in our screens. Labels on the right-hand side highlight the 10 genes showing the highest overall A3A essentiality. An extended heatmap showing all genes from certain DNA repair pathways is included in S7 Fig, and numerical data for graphs in this figure are provided in S3 Data. A3A, APOBEC3A; LFC, log₂ fold change; LUAD, lung adenocarcinoma.

in these genetic screens. For example, *RAD9A* loss sensitizes to a variety of agents including gemcitabine, hydroxyurea, bleomycin, AZD6738 (ATR inhibitor), and others (Fig 3B). *MCM8*, *MCM9*, or *HROB* loss sensitized to MNNG, cisplatin, MMS, trabectedin, and

camptothecin (Fig 3B). Loss of the *DDX11* helicase or *MAD2L2* (also known as *REV7*; component of the Shieldin complex and accessory subunit of the error-prone DNA polymerase ζ) sensitized to a wide gamut of DNA damaging agents tested (Fig 3B) [61]. This suggests that these hits may be generally critical to stalled forks, rather than specific to A3A-mediated damage [6,60,62,63].

In contrast, *HMCES*, *UBA6*, and *ATXN7L3* appeared to have a more restricted pattern of sensitization to DNA damaging agents (Fig 3B), indicating that they may represent better targets for selective killing of APOBEC-expressing cancer cells. Of those, *HMCES* exhibited a distinctive pattern that did not cluster with the other top 50 hits in our screen (Fig 3B). *HMCES* loss sensitized to exposure to $KBrO_3$ and $H_2O_2$ (oxidizing agent), and ionizing radiation (IR), which generates oxidative base damage and DSBs, in previous data [60]. This is consistent with the occurrence of AP sites as repair intermediates of oxidatively damaged DNA and a role for HMCES in protecting such AP sites. Intriguingly, *HMCES* loss also sensitized to illudin-S and duocarmycin, alkylating drugs with incompletely understood mechanisms of action, but less so to other alkylators (Fig 3B, S8 Fig) [60]. Overall, this joint analysis of previous genetic screening data under DNA damaging conditions, together with our APOBEC screens, indicates that HMCES has a specialized, rather than a general role in protecting against DNA damage. Further, this suggests that inhibiting HMCES would be selective for treating tumors undergoing certain types of DNA damage, such as APOBEC-mediated cytosine deamination, or in combination with specific therapeutic strategies, such as radiotherapy that is widely used in cancer treatment.

## HMCES depletion sensitizes A3A-expressing cells

To test these possibilities, we depleted HMCES in multiple lung cancer cell line backgrounds by shRNA depletion or CRISPR/Cas-9 knockout. Efficient depletion of HMCES mRNA and protein levels by shRNA in either LXF-289 or NCI-H358 (Fig 4A), which was not used for the screening, enhanced sensitivity to A3A expression to different extents. Sensitivity in LXF-289 cells was apparent at early and late times (Fig 4B), consistent with screening results, and accompanied by an arrest in G2/M phase (Fig 4C) and increased levels of the γH2AX DNA damage marker (Fig 4D). NCI-H358 cells also showed increased sensitivity to A3A expression (Fig 4E). Together with the A549 screening data, these experiments further support that *HMCES* loss is more toxic to A3A-overexpressing cells in multiple genetic backgrounds.

We next asked whether the top hits in our A3A genetic screen were dependent on the activity of TP53, by comparing the derived A549$^{TP53-/-}$ cell line with its progenitor A549 that has a wild-type *TP53* status. Most of the top hits from the initial assay were not among the genes that differed depending on *TP53* status in A549. A prominent exception was *HMCES* (S5 Fig), which exhibits a stronger loss of fitness phenotype upon A3A expression in *TP53*$^{-/-}$ cells than in wild-type cells (we also noted some signal for *CDC23*; S5 Fig). As further support of this, in the statistical analysis of previous genetic screening data from Project Achilles (Fig 3A), we found that the association between the APOBEC mutational signatures and sensitivity to *HMCES* loss holds only for the *TP53*-mutant cell lines, but not for the *TP53* wild-type cell lines (S9 Fig). To further test the epistatic interaction between *TP53* and *HMCES* under A3A expression, we directly assessed survival using colony-forming assays in the A549 cell line pair following A3A expression and HMCES depletion. This showed that A549$^{TP53-/-}$ cells were more sensitive to A3A than A549 following HMCES depletion with shRNA (Fig 4F). This suggests that HMCES inhibition could be used to target A3A-expressing cells that have lost TP53, as is the case in many tumors, while sparing TP53 wild-type cells to a large extent.

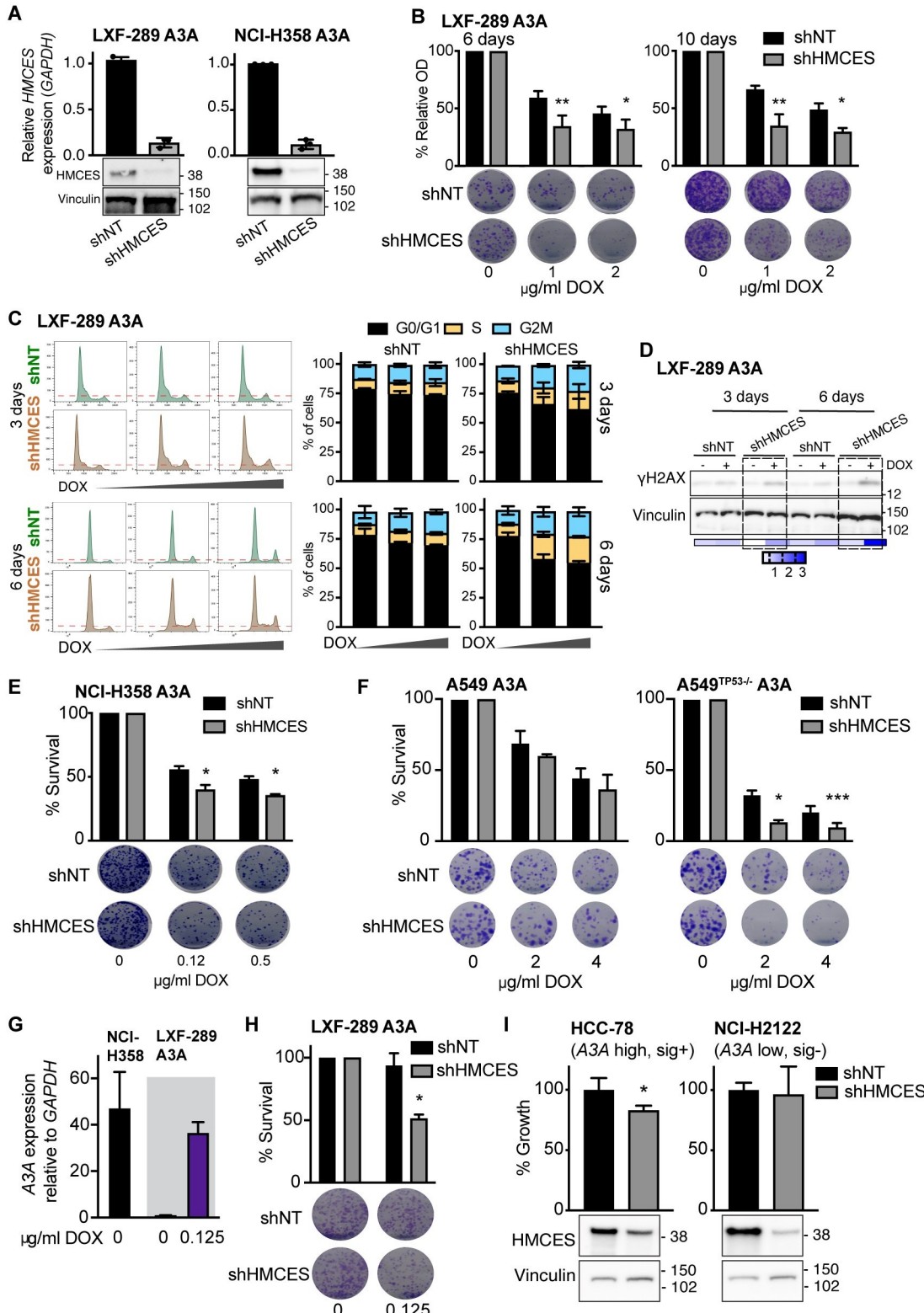

**Fig 4. Validation of the effects of HMCES depletion in multiple genetic backgrounds.** (**A**) HMCES levels in LXF-289 A3A and NCI-H358 A3A cell lines by qRT-PCR and western blot following transduction with shHMCES or a shNT. The qRT-PCR validation was repeated at least 3 times. Molecular mass is indicated in kilodaltons. (**B**) Reduction of HMCES sensitizes LXF-289 A3A cells to *A3A* expression in a growth assay for 6 and 10 days (repeated 3 times). (**C**) Representative histograms of cell cycle

progression (left panels) and quantitative analysis of LXF-289 A3A shHMCES and shNT (right panels). Cells were treated with DOX (0, 1, or 2 ug/ml) and harvested after 3 and 6 days (repeated 2 times). (**D**) Western blot of H2AX-S139 phosphorylation (γH2AX) in LXF-289 shHMCES and shNT cells after 3–6 days of A3A expression. Relative phosphorylation (0–3) was calculated normalizing the band densities of γH2AX to total Vinculin signal. Molecular mass is indicated in kilodaltons. (**E**) Reduction of HMCES sensitizes NCI-H358 A3A cells to *A3A* expression in a clonogenic survival assay after 15 days (repeated 2 times). (**F**) The effect of HMCES depletion is TP53 dependent. Clonogenic survival assays of A549 A3A or A549$^{TP53-/-}$ A3A cells are shown 10 days after treatment with the indicated dose of DOX (repeated 3 times). (**G**) *A3A* mRNA expression levels in LXF-289 A3A (0 and 0.125 ug/ml of DOX) and the parental NCI-H358 cell line relative to *GAPDH* measured by qRT-PCR (repeated 3 times). (**H**) Depletion of HMCES sensitizes LXF-289 A3A cells to *A3A* expression following low levels of DOX treatment in a clonogenic survival assay (repeated 3 times). (**I**) Growth inhibition (percentage of growth rate) measured with AlamarBlue for HCC-78 (A3A expressing, A3 mutational signature positive) and NCI-H2122 (A3A low, A3 mutational signature negative) cell lines transduced with shNT or shHMCES (repeated 3 times). HMCES levels are shown, and Vinculin is used as a loading control (bottom panels). Statistical analysis of shHMCES vs. shNT in all panels was performed using a 1-tailed unpaired *t* test; mean and SD are shown for all graphs. *, $p \leq 0.05$, **, $p \leq 0.01$, ***, $p \leq 0.001$. Uncropped western blots are provided in S1 Raw Images, and numerical data for all graphs are provided in S4 Data. A3A, APOBEC3A; DOX, doxycycline; qRT-PCR, quantitative real-time PCR; SD, standard deviation; shNT, nontargeting shRNA.

Given that our experimental models rely on the inducible expression of exogenous *A3A*, we wanted to ascertain the relevance of these results to endogenous A3A expression levels in cancer cells. Examination of endogenous *A3A* expression by quantitative real-time PCR (qRT-PCR; Fig 4G) indicated that LXF-289 cells do not express detectable levels of *A3A*, while NCI-H358 cells do. We titrated DOX levels down to achieve an induction of *A3A* mRNA levels in LXF-289 cells similar to that of endogenous *A3A* that we observed in NCI-H358 cells (Fig 4G). We then examined the effect on cell growth and found that this impaired the growth of LXF-289 when HMCES was depleted (Fig 4H), consistent with experiments using higher DOX levels (Fig 4B). To further address the issue of applicability of HMCES inhibition to endogenous *A3A* levels, we examined public data for gene expression and mutational signatures in other NSCLC cell lines, highlighting 2 contrasting examples: HCC-78, which express *A3A* and exhibit APOBEC mutational signatures SBS2 and SBS13 (S10 Fig) [64,65], and NCI-H2122, which do not express detectable *A3A* nor exhibit the A3 mutational signatures (S10 Fig). We confirmed their relative *A3A* expression by qRT-PCR (S10 Fig) and depleted HMCES using shRNA. While both cell lines showed reduced levels of HMCES protein, only the naturally A3A-expressing HCC-78, but not the A3A non-expressing NCI-H2122, showed significant defects in cell growth upon HMCES depletion (Fig 4I). Together, these data further support a role for HMCES in tolerating endogenous *A3A* expression in cancer cells.

## Sensitivity of A3A-expressing cells to HMCES loss is enhanced by DNA damage

In addition to sensitizing to A3A, previous data implicated HMCES in the sensitivity to a limited number of DNA damaging agents that included IR and KBrO$_3$ [42,60]. Considering this, and that DNA repair factors involved in multiple DSB repair pathways were identified in our A3A screens (Fig 2E), we examined the effects of combinatorial treatments on A3A-mediated toxicity. Survival of LXF-289 A3A cells was analyzed with or without DOX in combination with IR or KBrO$_3$ treatment. Both agents showed increased toxicity in cells expressing A3A (Fig 5A; $p \leq 0.05$ for synergistic activity, by *t* test against a Bliss independence baseline) [66]. We next examined the relative importance of the 3 PI-3 kinase-like kinase (PIKKs), ATM, ATR and DNA-PKcs, which regulate many of the individual proteins identified in the screens. LXF-289 cells were treated with DOX to induce A3A, and each of the PIKKs was inhibited using small molecule inhibitors [67]. As previously reported, we saw that the toxicity of A3A overexpression was strongly enhanced following treatment with ATR inhibitors [32,33]. In

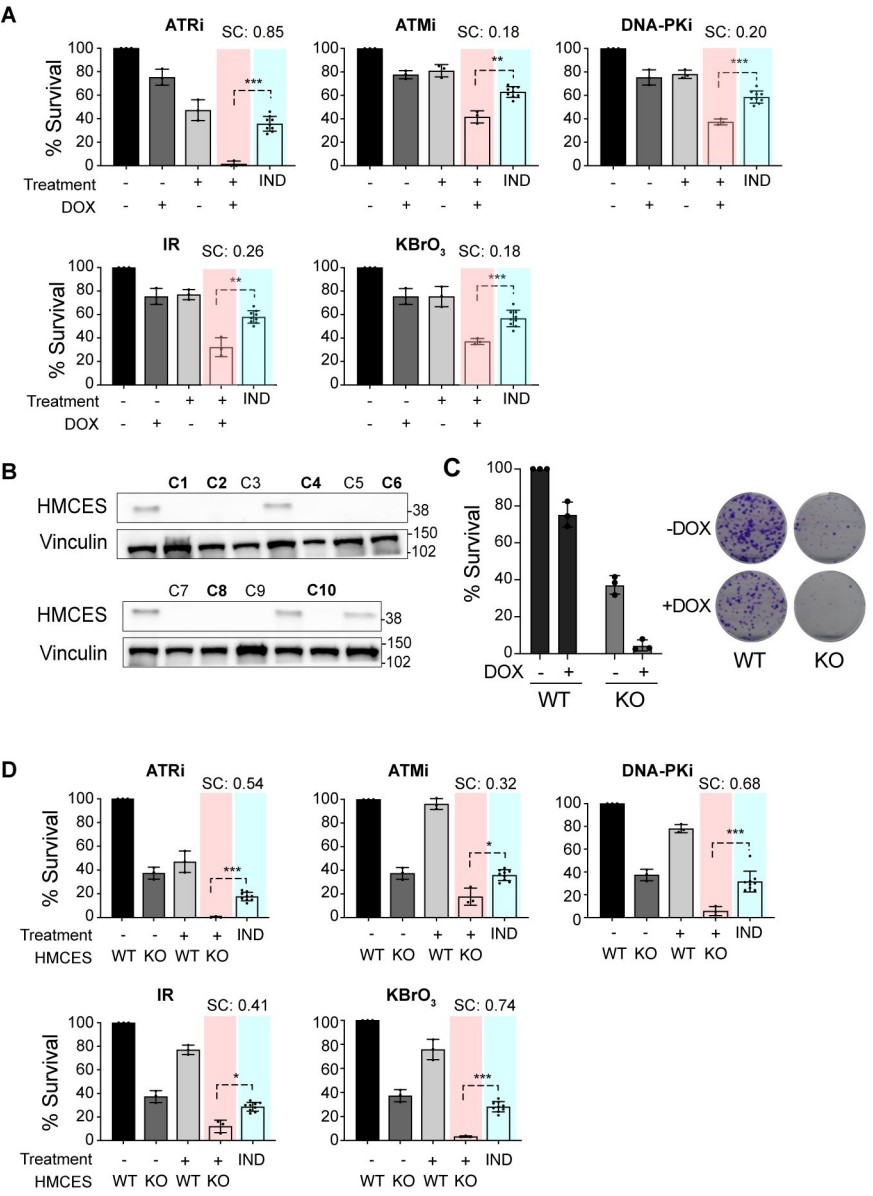

**Fig 5. A3A expression sensitizes to DNA damage and HMCES loss.** (**A**) Clonogenic survival measured by the colony formation assay in LXF-289 A3A cells after exposure to the indicated small molecule inhibitor, IR (5 Gy), or treatment with 0.1 mM KBrO$_3$ with or without DOX (0.125 ug/ml) to induce *A3A* expression. (**B**) HMCES western blot of lysates from LXF-289 A3A HMCES WT and KO clones. Vinculin was used as a protein loading control, and molecular mass is indicated in kilodaltons. (**C**) Clonogenic survival assay of LXF-289 A3A HMCES WT and KO cells upon overexpression of A3A by DOX. (**D**) Clonogenic survival assay comparing HMCES WT and KO cells after treatment with the indicated small molecule inhibitor, exposure to IR (5 Gy), or treatment with 0.1 mM KBrO$_3$ with or without DOX to induce A3A expression. For panels A and D, the "IND" column shows a Bliss independence model of additive activity of the 2 treatments, against which the combined treatment is tested (using *t* test, 2-tailed) to estimate synergistic activity [66]. Mean and SD are shown in all graphs. *, $p \leq 0.1$, **, $p \leq 0.05$ ***, $p \leq 0.01$. Each experiment was repeated 3 times, and primary numerical data are provided in S5 Data. A3A, APOBEC3A; DOX, doxycycline; IR, ionizing radiation; KO, knockout; SC, synergy score; SD, standard deviation; WT, wild-type.

addition, we saw that inhibitors of ATM and DNA-PKcs, which play key roles in DSB repair, led to synergistic cell killing with A3A overexpression ($p \leq 0.05$), albeit to a more modest extent than with the ATR inhibitor or with IR (Fig 5A).

We next examined the influence of HMCES on combination treatments by generating knockout cell lines (HMCES KO) in the LXF-289 A3A background using CRISPR/Cas-9 and single-cell isolation (Fig 5B). Six clones with no detectable HMCES protein levels (Fig 5B; bold type) were pooled to generate an HMCES KO cell culture for subsequent analysis. HMCES KO impaired the colony-forming capacity of LXF-289 A3A cells, and this was further reduced upon expression of A3A following DOX treatment (Fig 5C). HMCES KO cells also showed hypersensitivity to the inhibition of any of the PIKKs, in the absence of exogenous DNA damaging agents, as well as to treatment with IR or KBRO$_3$ (Fig 5D). Together, our data point to HMCES as an important dependency of cancer cells for survival and suggest that this can be exploited using combination therapies.

## Screening HMCES deficient cells reveals additional modifiers of the A3A response

As HMCES KO cells could still tolerate some A3A expression, we performed a secondary CRISPR/Cas-9 genome-wide screen to identify mediators of survival that were specific for LXF-289 A3A HMCES KO cells (Fig 6A). Positively selected genes in this assay are those whose deletion lessens the fitness penalty due to loss of *HMCES* upon *A3A* overexpression (alleviating epistasis), while negatively selected genes are those whose deletion increases this fitness penalty (aggravating epistasis). Expectedly, this screen identified a strong positive selection for the loss of *A3A*, presumably by targeting our inducible gene. Further, it identified *UNG*, the gene encoding the primary uracil-DNA glycosylases UNG1 and UNG2, which localize to the mitochondria and nucleus, respectively. This indicated that preventing the generation of AP sites by A3A conferred survival to HMCES KO cells, consistent with previous work [22,49]. In addition, the loss of multiple genes encoding subunits of the Mediator complex (*CCNC*, *MED25*), splicing regulators (*SCAF1*, *SCAF8*), the *FBXW7* E3 Ubiquitin ligase (a common tumor suppressor gene), the MMR protein *MSH2*, the Protein phosphatase 4 subunit *SMEK1* (PPP4R3A) that dephosphorylates γH2AX[68], the *ATF2* transcription factor, *CARM1* (PRMT4), and *FADD* (FAS-associated death domain protein) were among high-scoring hits that were positively selected in HMCES KO cells (S6 Table).

Among negatively selected genes in HMCES KO cells, 2 PIKK targets, *TDP1* (tyrosyl-DNA phosphodiesterase 1) and *VCPIP1*, were among the few genes implicated in DNA repair [72–75]. In addition, the gene encoding the Protein phosphatase 2A subunit (*PPP2R2A*), a candidate tumor suppressor that negatively regulates ATM-CHK2, was negatively selected [76,77].

Together, these data indicate that the sensitivity of HMCES null cells can be mitigated by the loss of *UNG*, implicating AP site generation in the toxicity, and that TDP1 and VCPIP1 may represent key backup activities for the resolution of A3A-mediated damage to promote cell survival.

## Discussion

Our results establish that HMCES is a key mediator of A3A toxicity in cancer cells. Given the increased levels of γH2AX DNA damage signaling observed in HMCES knockdown cells, as well as the enrichment in DSB repair factors in our screens, our results suggest that DSBs may be a major driver of A3A toxicity. Notably absent in our primary screens were factors involved in the BER pathway that would normally resolve AP sites and promote the base changes that are evident in the APOBEC mutational signature. We interpret this to mean that A3A lesions are likely not toxic outside of S-phase, where they can be repaired by BER and likely additional pathways.

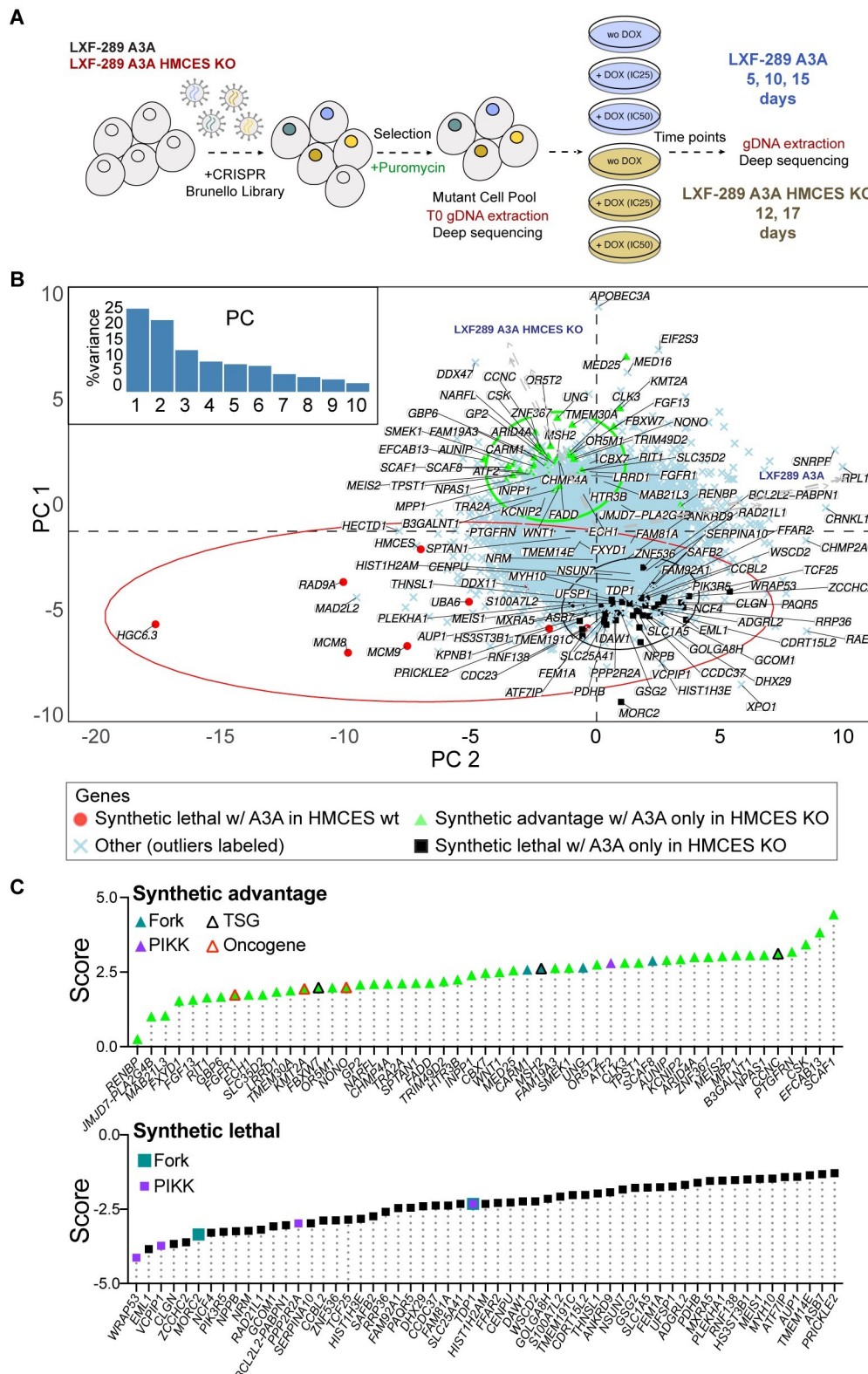

**Fig 6. Secondary screening identifies modifiers of the response to A3A in HMCES KO LXF-289 cells.** (**A**) Schematic of the secondary screen in LXF-289 A3A HMCES KO cells. (**B**) PC analysis of A3A conditional LFC scores for all genes across all 10 experimental conditions (see labels next to arrows, which show loadings of the conditions on

PC1 and PC2; HMCES wt refers to the parental LXF-289 cell line). Top genes, defined as the top 10 genes from S3C Fig, plus those that pass at least 1 filter (specified in the legend of S11 Fig), as well as additional genes that pass the filters when they are relaxed, are highlighted on the figure for each indicated comparison: green, synthetic advantage; black, synthetic lethality; red synthetic lethal in HMCES wt. Gene-level LFCs and GO analysis results are included in S6 and S7 Tables, and additional epistasis analysis is included in S11 Fig. (**C**) Selected genes from panel **B**, sorted by a score calculated as the mean of the standardized sgRNA LFCs and standardized MLE beta score differences from the 4 DOX vs. control comparisons of HMCES KO samples, minus the mean obtained in the same way for the HMCES wt samples. A negative score (black squares, as in **B**) indicates likely synthetic lethality with A3A expression in HMCES KO but not in HMCES wt, and a positive score (green triangles, as in **B**) suggests that the gene confers a synthetic advantage with A3A expression in HMCES KO, but not in HMCES wt (e.g., *UNG*). Known TSGs or oncogenes are indicated (see key, data derived from COSMIC: https://cancer.sanger.ac.uk/census). Gene products that are known PIKK substrates or regulators (PIKK) and those that have been demonstrated to localize to active replication forks (Fork) are indicated [69–71]. Numerical data for panels **B** and **C** are provided in S6 Data. A3A, APOBEC3A; gDNA, guide DNA; GO, Gene Ontology; KO, knockout; LFC, $\log_2$ fold change; PC, principal component; PIKK, PI-3 kinase-like kinase; sgRNA, single gRNA; TSG, tumor supressor gene; wt, wild-type.

This proposition is consistent with recently published work demonstrating that HMCES depletion sensitizes both immortalized human cells and cancer cells to A3A expression [49]. In addition, the study showed that HMCES loss reduced replication fork elongation in a manner dependent on the UNG-mediated production of AP sites, using a uracil-DNA glycosylase inhibitor. UNG2 depletion was also previously shown to suppress the accumulation of replication fork-associated AP sites and DSBs in ATRi-treated A3A-expressing cells [33]. Curiously, UNG2 depletion caused lethality in A3B-expressing cells through a mechanism dependent on MMR and TP53 [78]. We did not identify any glycosylases as sensitizers or suppressors of A3A-mediated toxicity in our primary screens, regardless of TP53 status. However, our secondary screen in HMCES KO cell lines identified UNG as a major positively selected hit in cells expressing A3A, reinforcing the proposition that AP site production underlies the sensitivity of cells to A3A expression [42,49]. It also suggests a mechanistic divergence between the toxicity of A3A and A3B expression.

In addition, our secondary screens identified the MMR protein MSH2 as positively selected, consistent with our recent data suggesting that A3A-mediated mutagenesis is mediated by MMR activity, resulting in the characteristic omikli pattern of clustered mutations [7]. In contrast, UNG, MSH2, or other MMR proteins were not identified as influencers of the survival of HMCES KO cells in the absence of A3A or DNA damage in our screens or in recent screens performed by others in HMCES KO cells in a HEK-293 background [79].

Replication fork slowing following A3A expression was shown to be due in part to the recruitment of TLS polymerases, particularly POLζ, as well as increased accessibility to APE1 endonuclease activity that is likely the source of DSBs [49]. While we did not identify APE1, potentially due to redundancy with other activities, we did find the POLζ subunit MAD2L2 as an A3A sensitizer (Fig 2C–2E). MAD2L2 has additional POLζ independent functions as an inhibitor of 5′-resection that promotes NHEJ-mediated repair, but we did not identify a strong dependence on other NHEJ factors in our primary screens [61]. Notably, this was specific to the LXF-289 cancer cell line, suggesting that the genetic background may have an appreciable impact on replication fork protection and stability, thus influencing how cells respond to AP sites at replication forks.

Aside from *HMCES*, the overall agreement between cell lines at the individual gene level was limited in our A3A screens. A previous analysis of essential genes using a DNA repair library in HEK-293 HMCES KO cells identified numerous proteins involved in HR and TLS, suggesting that loss of HMCES was synthetically lethal with the attenuation of these repair pathways [79]. In the absence of A3A expression, we observed limited overlap with that study when compared to our data in LXF-289 HMCES KO cells (S12 Fig). The ribonucleotide

reductase subunit RRM1, clamp loader subunit RFC3, and the HR factor XRCC3 were the main overlapping hits associated with DNA repair among negatively selected genes. This further highlights the likelihood that the individual genetic or epigenetic status of particular cell lines may play a significant role in the response to A3A expression, as well as to HMCES loss, consistent with the variable levels of cell cycle arrest caused by A3A expression that was cell line dependent (Fig 1). This may reflect the status of individual DNA repair pathways or DNA replication fork rates in the different genetic backgrounds. Despite the different phenotypic outcomes between cell lines, *HMCES* emerged as a common and prominent hit between the cell lines screened in our work (using an experimental system for A3A overexpression; Fig 2), as well as many other cell lines examined in the Project Achilles screens (using observational analysis of A3A mutational signatures in the cell line exomes; Fig 3) [80].

TP53 status clearly has a major influence on the genetic interaction between HMCES loss and A3A expression, implicating G1/S checkpoint status in the tolerance of A3A expression. TP53 status was shown to have a major influence on CRISPR/Cas-9 survival screens, and it is intuitive that checkpoint status would limit toxic DNA damage generated during S-phase and prevent mitotic catastrophe resulting from under-replicated DNA entering mitosis [81]. As A3A-mediated damage of ssDNA is S-phase specific, our results would again be consistent with the recently proposed model that HMCES shields AP sites in ssDNA from processing by BER endonucleases that would generate AP sites and toxic DSBs or from replication by TLS polymerases that would result in increased mutagenesis [27,42,43,49,79]. As the enzymatic activities of HMCES have been implicated in this function, accumulating data suggest that targeting HMCES would be an attractive strategy for the specific sensitization of A3A-expressing, TP53-deficient cancers.

Inhibition of the ATR-CHK1 kinases, which activate cell cycle checkpoints in S and G2 in response to replication stress, was shown to enhance A3A-mediated toxicity in AML, lung, ovarian, and osteosarcoma cell lines [32,33]. We did not identify either kinase in our primary screens; however, both genes are common essential genes in a large panel of cancer cell lines tested (DepMap.org), likely making them more difficult to detect as conditionally essential genes. Acute inhibition of ATR or CHK1 is better tolerated than genetic deletion or kinase dead alleles, which result in embryonic lethality, suggesting that it may be a more sensitive approach [82–85]. We did identify ATR as a hit in our LXF-289 HMCES KO cells that were not treated with DOX (no A3A) compared to parental cells (S12 Fig). This would potentially remove these cells from the pool early due to the HMCES KO interaction, thus precluding the identification of ATR following A3A treatment in those cells. Kinase dead alleles of both ATR and CHK1 have also been demonstrated to elicit phenotypes distinct from genetic deletion [82–85]. This suggests the possibility that the interaction between A3A and ATR/CHK1 inhibitors could be due to specific effects of the inhibitors, such as increased ssDNA stability or exposure. Regardless, our results further extend the robustness of previous observations to additional lung cancer cell lines, as we observed that A3A expression provoked particularly strong sensitivity to ATR inhibitors, among the interventions tested (Fig 5A) [32,33].

In contrast to previous work, we also found increased sensitivity to ATM and DNA-PKcs inhibitors in LXF-289 NSCLCs lacking HMCES, again likely reflecting differences in the genetic backgrounds analyzed [78]. While neither kinase is generally essential like ATR, and neither was identified as a strong hit in the screening, multiple PIKK substrates and regulators were identified, supporting the potential roles of these signaling pathways in survival (Figs 2E and 6C). As inhibitors for the PIKKs are in multiple clinical trials, targeting HMCES may represent a strategy to enhance their efficacy, particularly in combination with radiotherapy. This may be particularly potent in cancers with *PPP2R2A* deletions [77]. PPP2R2A is a negative regulator of ATM-CHK2 and a candidate tumor suppressor gene commonly deleted in ovarian,

prostate, liver, and bladder cancers [76,77,86]. Loss of PPP2R2A enhanced toxicity of A3A in HMCES KO cells (Fig 6B and 6C), and depletion or PPP2R2A was shown to enhance the toxicity of ATR-CHK1 or PARP inhibitors in NSCLC [86,87].

Of the secondary screen hits that were specific to HMCES KO cells expressing A3A, TDP1 stood out as the primary negatively selected DNA repair protein (Fig 6B and 6C). TDP1 plays a key role in resolving blocked 3′-ends, including Topoisomerase I (TOP1) DPCs [73]. The processing of TOP1-DPCs by TDP1 is preceded by the proteolytic degradation of TOP1 by the SPRTN and VCP/p97 proteases [88]. Following its activation by ATM/ATR, VCPIP1, another strong hit in the secondary screen, deubiquitinates the SPRTN metalloprotease to promote its activation and the removal of DNA–protein crosslinks (DPCs) that can cause damage during DNA replication [75,89]. TDP1 can then resolve the digested 3′-TOP1-DPC to facilitate DNA repair. TDP1 has also been shown to act as an AP-lyase and is essential in cells lacking APE1, which promotes the repair of AP sites by BER, indicating that it may play a wider role in the repair of A3A-mediated damage [74,90,91]. TDP1 inhibitors are currently being explored in clinical trials to sensitize cells to topoisomerase inhibitors and could be also used in conjunction with HMCES depletion/inhibition to sensitize cancer cells to APOBEC3 expression and perhaps other agents that generate AP sites [92].

In addition to sensitizing to A3A expression, HMCES deficiency also sensitized cells to treatment with IR and KBrO$_3$ treatment (Figs 3 and 5) [42,60]. As previously discussed, HMCES depletion sensitizes to a very limited spectrum of damaging agents compared to other hits in our screen that play more general roles in damage tolerance. The observation that additional hits, namely the poorly characterized SAGA complex component ATXN7L3 and ubiquitin-activating enzyme UBA6, shared a similar limited profile of damage sensitivity, similar to HMCES, suggests that a more definitive characterization of their roles is warranted in future work.

Collectively, existing data suggest that inhibition of HMCES is a promising strategy to suppress the APOBEC overexpressing, hypermutating tumor cell population, thereby slowing down the accumulation of genetic heterogeneity and preventing acquisition of new driver mutations or drug resistance mutations. Moreover, HMCES inhibition could augment the use of radiotherapy, which is widely used in the treatment of many cancer types, and enhance the effectiveness of small molecule inhibitors for DNA damage signaling kinases and repair enzymes that are currently being developed and tested in clinical trials.

## Methods

### Cell culture and generation of doxycycline-induced lung adenocarcinoma cell lines

LXF-289, NCI-H358, HCC-78, NCI-H2122, and A549 cell lines were purchased from the American Type Culture Collection (ATCC, United States of America) and the Deutsche Sammlung von Mikroorganismen und Zellkulturen GmbH (DSMZ, Germany) and maintained with RPMI-1640 or DMEM medium and supplemented with 10% fetal bovine serum and 5% penicillin-streptomycin. The DOX-inducible HA-tagged A3A plasmid (pSLIK-Neo A3A) was a kind gift from the Weitzman Lab [27]. Lentiviral particles were generated by transfection of HEK-293T cells. After transduction with the pSLIK-A3A lentivirus, LUAD cells were selected in 1 mg/mL Geneticin (Ibian Technologies, Spain).

### Western blotting

Total protein was extracted using 2xSDS buffer (4% SDS, 120 mM Tris-Cl pH 6.8, 20% Glycerol), 1x protease (Roche, Switzerland), and phosphatase inhibitors (MilliporeSigma,

Germany). Lysates were then sonicated at medium-high intensity (M2) for 15 min (Bioruptor, Diagenode, Belgium) and boiled for 10 min at 93˚C. The protein concentration was quantified using the DC Protein Assay (Bio-Rad, USA) kit. A total of 40 ug of proteins per sample were separated by SDS-PAGE using the 12% Mini-PROTEAN TGX Precast Protein Gels (Bio-Rad) and transferred to PVDF (Bio-Rad) using the 1.5 mm Gel pre-programmed protocol of the Trans-Blot Turbo system (Bio-Rad). Membranes were blocked with 5% milk/PBST for 1 h at RT and incubated overnight at 4˚C with primary antibodies against p53 (1:1,000; sc-47698, Santa Cruz Biotechnology, USA), Vinculin (1:5,000; V9264, MilliporeSigma), HA (1:4,000; AB9110, Abcam, United Kingdom), and HMCES (1:500; HPA044968, Atlas Antibodies, MilliporeSigma). Bands were detected with the corresponding HRP-conjugated secondary antibodies for 1 h at RT and visualized using the ECL-Plus kit (GE Healthcare, USA).

## Growth arrest and colony formation assays

For growth assays, cells were plated at density of 1,000 cells/well in a 96-well plate. After 24 h, cells were treated with increasing doses of DOX (0 to 8 ug/ml) and cultured for 72 h. Alamar-Blue reagent (Thermo Fisher Scientific, USA) was added to cell culture media 4 h prior to reading fluorescence with a SYNERGY H1M fluorescence plate reader (BioTek, USA). For the colony formation assays, between 250 and 1,000 cells per well were plated in a 12-well plate. Colonies were fixed with formalin (MilliporeSigma) and stained with a 0.01% crystal violet (MilliporeSigma) solution in 20% methanol. For some cell lines, quantification was performed by reading absorbance at 590 nm after the addition of 10% acetic acid.

## Drug sensitivity assays

Colony-forming assays for drug sensitivity testing were performed by plating the cells at a density of 500 cells/well in a 6 well-plate, in triplicate. Moreover, 24 h after plating, the following drug treatments were used: DNA-PKi (KU57788) 1 uM (MedChemExpress, USA), ATMi (KU55933) 5 uM (MilliporeSigma), ATRi (AZD6738) 0.5 uM (MedChemExpress), and $KBrO_3$ 0.1 mM (MilliporeSigma). IR (5 Gy) was administered using a Maxishot.200 X-Ray cabinet (Krautkramer Forster, Spain). For the induction of A3A expression, DOX was added at a concentration of 0.125 ug/ml ($IC_{25}$). The drug treatment was maintained in the growth media for the duration of the experiment (10 days), after which cells were fixed and stained with crystal violet. The number of colonies was quantified with Fiji (ImageJ, National Institutes of Health, USA). Colony-forming capacity is presented as a percentage of the vehicle-treated (DMSO 0.025%) control.

## Cell cycle analysis

Cells were fixed in 70% ethanol for at least 2 h at −20˚C and resuspended in a PBS solution containing 35 ug/ml propidium iodide (MilliporeSigma) and 100 ug/ml RNAseA (Roche). Between 5,000 and 10,000 cells were analyzed per sample. Data were acquired on a Gallios A94303 Flow Cytometer (Beckman Coulter, USA) in the Cytometry Core Facility of the University of Barcelona and analyzed by FlowJo software (BD, USA).

## RNA extraction, cDNA synthesis, and qRT-PCR

RNA extraction was performed using the Maxwell 16 LEV simplyRNA cell Kit (Promega, USA) according to manufacturer's instructions. cDNA synthesis was performed with the high-capacity cDNA reverse transcription kit (Life Technologies, USA). qRT-PCR was performed with SYBR Select Master Mix for CFX (Applied Biosystems, USA) or TaqMan universal PCR

Master Mix II (Applied Biosystems) on a StepOnePlus Real-time PCR System (Applied Biosystems). Probes and primers are shown in S8 Table.

## Generation of stable KO and knockdown cells

For A549 ± A3A and LXF-289 ± A3A cell lines, the NickaseNinja (ATUM, USA) vector co-expressing 2 gRNAs (pD1401-AD: CMV-Cas9N-2A-GFP, Cas9-ElecD) was used to generate the TP53 KO and the HMCES KO cells. TP53 gRNA sequences (GCAGTCACAGCACATG ACGG) (GATGGCCATGGCGCGGACGC) and HMCES gRNA sequences (CAGTGA ATGGATCTCTACAA) (GAGCTTGCGCCTACCAGGAT) were designed using the ATUM gRNA Design Tool. Moreover, 48 h post-transduction, positive GFP cells were sorted by FACS (FACSAria Fusion, BD, USA) and plated into 96-well plates. After 15 days, clones were collected and validated by western blot using the following primary antibodies: p53 (1:1,000; sc-47698, Santa Cruz Biotechnology); Vinculin (1:5,000; V9264, MilliporeSigma); and HMCES (1:500; HPA044968, Atlas Antibodies). The HMCES knockdown stable cell lines (HCC-78, NCI-H2122, LXF-289 A3A, NCI-H358 A3A, A549 A3A, and A549$^{TP53-/-}$ A3A) were made using the Mission shRNA lentiviral vector NM_020187.1-133s1c1 (MilliporeSigma). Lentiviral particles were produced in HEK-293T cells using a pLKO.1-shRNA plasmid. The cell lines were transduced and selected with puromycin for 72 h. As a control, we transduced LXF-289 (A3A) cells with the nonmammalian shRNA Control Plasmid DNA shC002 (MilliporeSigma).

## CRISPR/Cas-9 screening

For sgRNA screening of the A549 ± A3A, A549$^{TP53-/-}$ ± A3A, LXF-289 ± A3A, cells were infected with the Brunello CRISPR Knockout Pooled Library (73179-LV, Addgene, USA). Infection with lentiviruses was performed at an MOI ≤0.4 for all cell lines. At 24 h postinfection, the medium was replaced with a selection medium containing puromycin (2 ug/mL). After 5 to 6 days of selection, cells were split into the different experimental conditions: For LXF-289 cell line, without and with DOX (0.125 and 2 ug/ml corresponding to IC$_{25}$ and IC$_{50}$, respectively). For LXF-289 HMCES KO secondary screening, without and with DOX (0.03 and 0.125 ug/ml corresponding to IC$_{25}$ and IC$_{50}$, respectively). For A549 cell line, without and with DOX (3.9 ug/ml). All cell lines were passaged every 3 days (up to 15 days), and for each time point, the number of cells needed to maintain the predetermined coverage of 400- to 500-fold was taken. DNA extraction was performed using the DNA genomic Kit (Puregene Cell and Tissue Kit, Qiagen, Germany).

## NGS library preparation and sequencing

Next-generation sequencing (NGS) libraries were prepared by 2-step PCR: For the first one, a total of 20 ug of DNA per a 12X reaction was used, and for the second PCR, a set of primers harboring Illumina TruSeq adapters as well as the barcodes for multiplexing were used (for all primers used, see S8 Table). Sequencing was carried out in the CNAG sequencing unit using 6 lanes of a 1x50 HiSeq.

## Statistics

The statistical analyses were performed using Prism software (GraphPad, USA) version 8.0. Each functional experiment was repeated 2 times or 3 times (as specified in the figure or legend). Differences between groups were analyzed by Student $t$ test assuming unequal variances.

### Independent validation

We downloaded mutational signatures for the cell lines from Petljak and colleagues [65] and gene essentiality fitness score from Project Achilles [58]. We selected the cell lines from HNSC, LUAD, and LUSC. For the top 10 scoring genes in our analysis, we fitted a linear regression model between the cell lines fitness score and the signature loadings for signatures SBS2, SBS13, and SBS2+SBS13 (APOBEC signatures). We compared the slope and $p$-values obtained. The $p$-value is obtained from a $t$ test (1-tailed lower).

### In silico analysis of DNA damage sensitivity to DNA damaging agents

We downloaded the previously published data for the Z-scores after genotoxin exposure screens [60]. We compared how our top 50 genes (essential upon A3A overexpression) versus 50 genes that are not essential in our screens behaved after the genotoxin exposure.

### In silico analysis of CRISPR/Cas-9 screening results

For alignment of the generated reads to the library, read counting, read count normalization, QC analysis of the samples, and calculation of the sgRNA counts LFC, we used MAGeCK--VISPR [51]. For pairwise comparisons, we employed the robust rank aggregation (RRA) algorithm, using as treatment the DOX-induced A3A sample, for each cell line (A549, A549$^{TP53 -/-}$, and LXF-289) and time point.

Estimation of gene essentiality and sgRNA efficiency was achieved using the maximum likelihood estimation (MLE) algorithm provided by MAGeCK-VISPR. Namely, gene essentiality was estimated by comparison of the normalized sgRNA counts between each sample (A549 and A549$^{TP53-/-}$ time 9, 12, and 15, and LXF-289 time 5, 10, and 15) and its corresponding time 0 sample, which yielded a beta score per gene and sample. The beta score distribution for each sample was standardized by subtraction of the mean and division by the SD, and a final gene essentiality score was obtained by averaging the resulting Z-scores across samples.

Finally, we used the FluteMLE function from the R package "MAGeCKFlute" [93] for (i) normalization of the beta scores yielded by MAGeCK-VISPR MLE using a built-in set of 622 essential genes as a reference; and (ii) comparison of the essentialities between conditions (DOX-induced A3A versus control) within each cell line and time point, applying a significance cutoff of 2 SDs (S4 Fig). This allowed us to identify genes that were negatively selected in the A3A-expressing samples, but not selected in the control samples.

## Supporting information

**S1 Fig. Validation of the A549 ± A3A TP53−/− clones.** Western blot of the A549p53−/− (left panel) and the A549 A3A transduced p53−/− clones (right panel) generated using CRISPR/Cas-9 targeting. Uncropped blots are provided in S1 Raw Images. A3A, APOBEC3A. (TIF)

**S2 Fig. Effects of DOX treatment.** Volcano plot highlighting genes for which there could exist interaction with DOX per se. The x-axis represents the difference of the (normalized) MAGeCK-MLE's beta score between treating a sample with DOX (IC$_{25}$) and the corresponding control sample, averaged across the A3A plasmid-free version of all cell lines sampled after 15 days of cell culture. Intuitively, genes with significant negative beta score differences suggest conditional essentiality with DOX. The y-axis represents the $-\log_{10}$ FDR of the Fisher's combined $p$-value (either lower or upper tail) across samples. Numerical data used for the plot are provided in S2 Data. A3A, APOBEC3A; DOX, doxycycline; FDR, false discovery rate. (TIF)

**S3 Fig. Change in sgRNA count LFC dependent on days of cell culture before sampling or DOX dose.** LFC (y-axes) represents the cell count differences between a sample treated with DOX (IC$_{25}$ in plots **A**, **B**, and **C**) and the corresponding control (untreated) sample. (**A**) The top 4 genes are shown after sorting based on the overall score. The 4 sgRNAs targeting each gene are shown separately, and their count distribution is represented as a boxplot. Lines join the median sgRNA counts for each gene. One of the top 5 genes, HGC6.3, was excluded from the plot due to low data quality (see S1 Table). (**B**) The sgRNAs shown are the 1,000 nontargeting control sgRNAs in the Brunello library [50] and the top 100 genes after sorting genes based on the overall score. Lines join the median sgRNA counts for each distribution. Red dots indicate the LFC of sgRNAs for the HMCES gene. (**C**) The top 10 genes are shown after sorting by the overall score. Here, the HGC6.3 gene is included, while according to MAGeCK-MLE, it had low sgRNA efficiency (S1 Table), possibly causing the rather erratic trends across time points that it exhibits. (**D**) Difference in LFC dependent on DOX dose, either IC$_{25}$ or IC$_{50}$, in the LXF289 cell line. Columns show LFCs at different sampling times. The top 4 genes by overall score are shown. Numerical data used for all plots are provided in S3 Data. DOX, doxycycline; LFC, log$_2$ fold change; sgRNA, single gRNA.
(TIF)

**S4 Fig. Analysis of genetic screening data using an additional statistical methodology (MAGeCK-MLE).** (**A**) Out of a total of 339 genes identified as conditionally essential by MAGeCK-MLE in either of the 2 cell lines examined (see next point), this figure shows those genes that were significant in more than 1 sample (time point/cell line combination). A blue box indicates that the genes in that row are conditionally essential (under A3A overexpression) in the corresponding sample, while a gray box indicates that there is no significant essentiality. HMCES is the gene found to be conditionally essential in the highest number samples—all samples except the earliest time point of LXF-289, time 5. (**B**) MAGeCK-MLE visualizations ("nine-square plots" as in [93]) based on (left) A549 TP53−/− and (right) LXF-289 cell lines, both sampled at day 15. Points represent genes distributed according to the between-samples normalized beta scores (enrichments) for the control sample (untreated, x-axis) and DOX-treated sample (at IC$_{25}$ concentration, y-axis). Vertical and horizontal dotted lines indicate 2 SDs of the beta score distribution away from 0 to each side. Analogously, diagonal dotted lines represent 2 SDs of the distribution of between-treatment beta score differences away from 0 to each side. Therefore, genes located in the bottom center square ("Group4" genes) have MAGeCK beta scores different between the control and treated sample, being not different from 0 in the control (i.e., no evidence of selection) but significantly negative in the treatment (i.e., negatively selected); in other words, these genes are conditionally selected under A3A overexpression (DOX-induced). The top 10 genes are labeled in each square. HMCES is a top hit in both cell lines. Numerical data used for all plots are provided in S4 Data. A3A, APOBEC3A; DOX, doxycycline; SD, standard deviation.
(TIF)

**S5 Fig. Contrasts of A3A conditionally essential genes between different genetic backgrounds.** (**A**) Contrast between *TP53* backgrounds of the A549 cell line. Red circles are genes with a consistently strong negative LFC (below −0.4) in both *TP53* backgrounds (−/− above, wild-type below), considering the mean LFC after 9, 12, and 15 culture days (y-axes); LFC represents the cell count differences between a sample treated with DOX (IC$_{25}$) and the corresponding control (untreated) sample. The circle area shows the beta score (enrichment) calculated with MAGeCK-MLE and averaged across the time points: A more negative beta score indicates stronger gene essentiality irrespective of treatment. X-axes represent the

Human Protein Atlas consensus normalized (across cell lines) transcript expression levels (NX) in lung tissue for each gene. Among the hits, *HMCES* is prominent in the TP53−/− background but not in the wild-type background, has moderate expression levels in lung tissue, and does not appear to be generally strongly essential. (**B**) In an analogous manner, this plot shows the contrast of A3A conditionally essential genes between the A549[TP53−/−] and the LXF-289 genetic backgrounds; here, blue circles are genes with a consistently strong negative LFC (below -0.4) in both cell lines. Among the hits, *HMCES*, *RAD9A* and, to some extent, *MCM8* appear consistent in both backgrounds; of these 3 genes, *HMCES* has somewhat higher expression levels in lung tissue and is the least essential in these cell lines. *HMCES* is the only hit that is consistent in both comparisons, and this is noted by using a purple color to highlight it. Numerical data for all plots are provided in S5 Data. A3A, APOBEC3A; DOX, doxycycline; LFC, $\log_2$ fold change.
(TIF)

**S6 Fig. QC of sequencing reads.** (**A**) Number of total sequenced reads per sample. Light blue fraction represents the percentage of reads that are unequivocally unmapped to the library, which is below the recommended maximum of 35% in all samples [51]. (**B**) Number of library sgRNAs that have 0 counts per sample. Figures are higher in late samples, but this is to be expected due to negative selection. Overall, sgRNAs with 0 counts are <1% of total sgRNAs in Brunello library (approximately 77K) [50,51]. Namely, the maximum number of sgRNAs is 461. (**C**) Gini index of log-scaled read count distributions. This measure of the evenness across all sgRNA counts is below the recommended maximum of 0.2 in all samples [51]. Also, the Gini index is expected to increase in later time points. Data used for plots are provided in S6 Data. QC, quality control; sgRNA, single gRNA.
(TIF)

**S7 Fig. Differential fitness scores.** Differential fitness score (from project Achilles) upon APOBEC mutational signatures burden for the top 10 genes that are essential upon A3A overexpression in our screens (i.e., genes with the most negative mean LFC across 6 data points). Data used for the plots can be found in S5 Table. A3A, APOBEC3A; LFC, $\log_2$ fold change.
(TIF)

**S8 Fig. Many APOBEC-sensitizing genes, but not HMCES, also sensitize to a variety of other DNA damaging agents.** Left panel of heatmap shows a gene-level normalized LFC (gene essentiality score) upon A3A overexpression for 2 cell lines and for 3 time points (Biayna et al. screens); right panel shows Z-scores of gene essentiality after genotoxin exposure (Olivieri et al. screens) [60]. Data for 50 genes that are essential upon A3A overexpression in our screens (i.e., genes with the most negative mean LFC across 6 data points) (labeled "top"), and 521 DNA repair genes. Labels on the right-hand side highlight the 10 genes showing the highest overall A3A essentiality. Data used for the plots are provided in S7 Data. A3A, APOBEC3A; LFC, $\log_2$ fold change.
(TIF)

**S9 Fig. Gene essentiality fitness score from project Achilles vs. APOBEC mutational signatures exposures.** Cell lines originating from head and neck squamous cell carcinoma, LUAD, and lung squamous cell carcinoma were analyzed for the 4 genes with the greatest overall score in our genetic screens, while examining TP53 mutated (mut) and TP53 wt cell lines separately. The slope and *p*-value (1-tailed, lower) for the regression model for both APOBEC mutational signatures are shown within each panel. The more negative the slope, the more sensitive the cell lines are to the depletion of the particular gene at a higher level of the APOBEC mutational signature. Data used for the plots are provided in S7 Data. LUAD, lung adenocarcinoma; wt,

wild-type.
(TIF)

**S10 Fig. Endogenous expression and A3 mutational signature status of cell lines.** (**A**) Endogenous *A3A* mRNA expression levels in HCC-78 and NCI-H2122 cells relative to *GAPDH* measured by qRT-PCR (2 replicates). (**B**) A3A gene expression (TPMs) downloaded from EA (https://www.ebi.ac.uk/gxa/home) and (**C**) APOBEC mutational signatures (SBS2 and SBS13) burden downloaded from Petljak et al. and Jarvis et al. and normalized across cell lines (Z-score) [64,65]. Data used for the plots are provided in S7 Data. A3A, APOBEC3A; EA, expression atlas; qRT-PCR, quantitative real-time PCR.
(TIF)

**S11 Fig. Genes in epistasis with A3A expression in HMCES KO cells.** (**A**, **B**) Venn diagrams containing genes that are in epistasis with A3A expression (panel **A**, synthetic sickness/lethality, panel **B**, synthetic advantage) exclusively within an HMCES KO background, when applying 3 complementary statistical methodologies. Genes in the red circle have a standardized sgRNA LFC $<$-2 (**A**) or $>$2 (**B**) in the 4 DOX vs. control comparisons (IC$_{25}$-t12, IC$_{50}$-t12, IC$_{25}$-t17, and IC$_{50}$-t17) exclusively in HMCES KO samples. Genes in the blue circle fulfill the same criteria but using the MAGeCK-MLE standardized beta score difference, instead of the LFC. Lastly, the yellow circle contains genes whose normalized beta scores are not different from 0 in the control sample while they are significantly different from 0 (**A**, lower; **B**, higher) in the A3A-expressing sample, exclusively in an HMCES KO background: Specifically, this corresponds to the "bottom center" (A) or "top center" (B) square of MAGeCK-FLUTE's nine-square scatterplot visualization (see panel B of S4 Fig). Data used for the plots are provided in S7 Data. A3A, APOBEC3A; DOX, doxycycline; KO, knockout; LFC, log$_2$ fold change; sgRNA, single gRNA.
(TIF)

**S12 Fig. Genes differentially required for fitness between LXF-289 wt and LXF-289 HMCES KO cells.** Plot depicts the overall score of the top 65 targeted genes that reduced the viability of LXF-289 HMCES KO cells using the mean LFC from 2 time point comparisons (LXF-289 A3A wt t10 vs. LXF-289 A3A HMCES KO t12 and LXF-289 A3A wt t15 vs. LXF-289 A3A HMCES KO t17). Genes implicated in the DNA damage response are in bold, PIKK substrates/regulators or kinases are in red, and those in common with a previously published screen in HEK-293 based HMCES KO cells (considering those genes with a negative *p*-value $<$0.05) are in blue [79]. Full data set used for the plots is provided in S7 Data. A3A, APOBEC3A; KO, knockout; LFC, log$_2$ fold change; wt, wild-type.
(TIF)

**S1 Table. Estimated sgRNA efficiencies for the top 11 genes.** Efficiencies of sgRNAs were estimated by the MAGeCK-MLE algorithm in A549$^{TP53-/-}$ and LXF-289 cell lines. Informally, the efficiencies estimate the probability that a given sgRNA is able to generate an inactivating double-strand DNA break in the targeted gene. Only the *HGC6.3* gene shows overall low sgRNA efficiencies, particularly in the LXF-289 cell line, suggesting that the high log-fold change scores therein are an artefact (S3 Fig). *HMCES* has near-perfect sgRNA efficiencies. sgRNA, single gRNA.
(PDF)

**S2 Table. Gene-level data for all primary screens.**
(XLSX)

**S3 Table. Enriched GO Biological Process terms obtained from GOrilla.** GOrilla (http://cbl-gorilla.cs.technion.ac.il/) was used to analyze GO using a $p$-value threshold set at $10^{-3}$. Samples included are A549$^{\text{TP53}-/-}$ at time points 9, 12, and 15 days, and LXF-289 at time points 5, 10, and 15 days. GO, Gene Ontology.
(XLSX)

**S4 Table. AUC of each sample.** AUC per sample, based on the capacity of the CRISPR screening to discriminate between known sets of essential [93–95] and nonessential [94] genes by their normalized read counts. For comparison, the AUCs for the same overall sets of genes in the genetic screens (RPE1 cell line) from Brown et al. (2019) [54] have been included: Note that, while our screening was based on the Brunello library[50], Brown et al. employed the TKO library, so the gene overlap is not total. AUC, Area under the receiving operating characteristic curves.
(PDF)

**S5 Table. Table of differential fitness scores.** Differential fitness scores (from project Achilles) upon APOBEC mutational signatures (SBS2, SBS13, and SBS13+2) burden for the top 10 genes that are essential upon A3A overexpression in our screens (i.e., genes with the most negative mean LFC across 6 data points). A3A, APOBEC3A; LFC, $\log_2$ fold change.
(PDF)

**S6 Table. Gene-level data for the secondary genetic screen in HMCES KO cells.** KO, knockout.
(XLSX)

**S7 Table. GO enrichment analysis of the secondary screening data.** GOrilla (http://cbl-gorilla.cs.technion.ac.il/) was used to analyze GO of secondary screen results using a $p$-value threshold set at $10^{-3}$. GO, Gene Ontology.
(XLSX)

**S8 Table. Primers used in this study.** List of qRT-PCR primers (TaqMan/oligonucleotides) and PCR primers for library amplification and NGS. Primer sequence obtained from PrimerBank denoted with an $^{*}$ [96]. NGS, next-generation sequencing; qRT-PCR, quantitative real-time PCR.
(PDF)

**S1 Data. Numerical raw data.** All numerical raw data associated with Fig 1A and 1C. File contains multiple tabs with labels corresponding to the relevant figure.
(XLSX)

**S2 Data. Numerical raw data.** All numerical raw data associated with Fig 2B, 2C and 2E and S2 Fig. File contains multiple tabs with labels corresponding to the relevant figure.
(XLSX)

**S3 Data. Numerical raw data.** All numerical raw data associated with Fig 3A and 3B and S3A–S3D Fig. File contains multiple tabs with labels corresponding to the relevant figure.
(XLSX)

**S4 Data. Numerical raw data.** All numerical raw data associated with Fig 4A–4I and S4B Fig. File contains multiple tabs with labels corresponding to the relevant figure.
(XLSX)

**S5 Data. Numerical raw data.** All numerical raw data associated with Fig 5A, 5C and 5D and S5A and S5B Fig. File contains multiple tabs with labels corresponding to the relevant figure. (XLSX)

**S6 Data. Numerical raw data.** All numerical raw data associated with Fig 6B and 6C and S6 Fig. File contains multiple tabs with labels corresponding to the relevant figure. (XLSX)

**S7 Data. Numerical raw data.** All numerical raw data associated with S8–S12 Figs. File contains multiple tabs with labels corresponding to the relevant figure. (XLSX)

**S8 Data. Raw sgRNA count data.** Sample information for raw count sgRNA data in S9–S12 Data for genetic screens performed in this work. sgRNA, single gRNA. (XLSX)

**S9 Data. Raw sgRNA counts for A549p53KO.** KO, knockout; sgRNA, single gRNA; wt, wild-type. (XLSX)

**S10 Data. Raw sgRNA counts for A549 p53wt.** sgRNA, single gRNA; wt, wild-type. (XLSX)

**S11 Data. Raw sgRNA counts for LXF289 HMCESwt.** sgRNA, single gRNA; wt, wild-type. (XLSX)

**S12 Data. Raw sgRNA counts for LXF289 HMCES KO.** KO, knockout; sgRNA, single gRNA. (XLSX)

**S1 Raw Images. Uncropped western blots from all main and Supporting information figures.** (PDF)

## Acknowledgments

We thank members of the Stracker and Supek labs for input and discussions and M. Weitzman and A. Green for sharing reagents and unpublished data.

## Author Contributions

**Conceptualization:** Fran Supek, Travis H. Stracker.

**Data curation:** Miguel M. Álvarez, Marina Salvadores, Jose Espinosa-Carrasco, Fran Supek, Travis H. Stracker.

**Formal analysis:** Miguel M. Álvarez, Marina Salvadores, Jose Espinosa-Carrasco, Travis H. Stracker.

**Funding acquisition:** Fran Supek, Travis H. Stracker.

**Investigation:** Josep Biayna, Isabel Garcia-Cao, Miguel M. Álvarez, Marina Salvadores, Jose Espinosa-Carrasco, Marcel McCullough, Fran Supek, Travis H. Stracker.

**Methodology:** Josep Biayna, Miguel M. Álvarez, Marina Salvadores, Jose Espinosa-Carrasco, Fran Supek, Travis H. Stracker.

**Project administration:** Fran Supek, Travis H. Stracker.

**Resources:** Fran Supek, Travis H. Stracker.

**Supervision:** Fran Supek, Travis H. Stracker.

**Validation:** Josep Biayna, Isabel Garcia-Cao.

**Visualization:** Josep Biayna, Isabel Garcia-Cao, Travis H. Stracker.

**Writing – original draft:** Fran Supek, Travis H. Stracker.

**Writing – review & editing:** Josep Biayna, Isabel Garcia-Cao, Miguel M. Álvarez, Marina Salvadores, Jose Espinosa-Carrasco, Fran Supek, Travis H. Stracker.

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
