## [Editor Report · Decision Letter 0]

3 Aug 2020

Dear Dr Stracker, 

Thank you for submitting your manuscript entitled "Loss of HMCES is synthetic lethal with APOBEC activity in cancer cells" for consideration as a Research Article by PLOS Biology.

Your manuscript has now been evaluated by the PLOS Biology editorial staff as well as by an academic editor with relevant expertise and I am writing to let you know that we would like to send your submission out for external peer review.

Please re-submit your manuscript within two working days, i.e. by Aug 05 2020 11:59PM.

Kind regards,

Ines Alvarez-Garcia, PhD,

Senior Editor

PLOS Biology

---

## [Decision Letter · Decision Letter 1]

18 Sep 2020

Dear Dr Stracker,

Thank you very much for submitting your manuscript "Loss of HMCES is synthetic lethal with APOBEC activity in cancer cells" for consideration as a Research Article at PLOS Biology. Your manuscript has been evaluated by the PLOS Biology editors, an Academic Editor with relevant expertise, and by three independent reviewers.

We must apologise for the fact that a lapse in internal communication meant that although we considered this manuscript under our "anti-scooping" policy with respect to the related paper by Mehta et al (Cell Reports, June 2020), we unfortunately neglected to inform our reviewers of this policy. You will see that two of the reviewers mention this paper, and we will explain and apologise to them for this omission. According to our policy, we did not consider any issues of novelty raised by the reviewers with respect to Mehta et al.

You'll see that while reviewer #1's assessment is unfortunately coloured by our error, the other two reviewers are broadly positive about your study. Between them, however, the three reviewers raise significant concerns, some of which will need new experimental data to address them before we can consider the paper further.

In light of the reviews (below), we will not be able to accept the current version of the manuscript, but we would welcome re-submission of a much-revised version that takes into account the reviewers' comments. We cannot make any decision about publication until we have seen the revised manuscript and your response to the reviewers' comments. Your revised manuscript is also likely to be sent for further evaluation by the reviewers.

We expect to receive your revised manuscript within 3 months. 

**IMPORTANT - SUBMITTING YOUR REVISION**

*Re-submission Checklist*

*Published Peer Review*

*PLOS Data Policy*

*Blot and Gel Data Policy*

Sincerely,

Roli Roberts

Senior Editor,

rroberts@plos.org,

PLOS Biology

REVIEWERS' COMMENTS:

Reviewer #1:

Upregulation of APOBEC3A and APOBEC3B contribute to mutagenesis in certain types of cancer. Thus, finding vulnerabilities of tumours with high expression of A3A/A3B may open the door to novel targeted therapies. APOBEC3 activity induces abasic sites by cytosine deamination, leading to high levels of mutations and replicative stress. However, the genes and pathways involved in the repair of DNA damage induced by A3A/A3B overexpression are not yet known. Recent works have shown that HMCES is a DNA repair factor which binds covalently to abasic sites induced by exogenous oxidative damage, promoting microhomology-mediated end joining and avoiding mutagenic pathways during replication. 

In this work, Byana and colleagues employed an unbiased functional genomic approach to find genetic vulnerabilities using two A549 and LXF-289 lung adenocarcinoma cell lines conditionally expressing APOBEC3A, putting forward a set of target proteins. The screening approach is well executed, obtaining a dataset enriched in DNA repair genes, as expected. They focused on HMCES, providing experimental evidence that cells lacking functional levels of HMCES are hypersensitive to A3A overexpression, accumulate DNA damage with defects in cell cycle progression. They finally proposed that HMCES is an attractive target for p53-mutated/APOBEC3A overexpressing tumors, demonstrating that the efficacy of this approach synergizes in combination with other previously proposed therapeutic approaches such as ATRi inhibition.

The main problem with this manuscript is the fact that the vulnerability of APOBEC3A-overexpressing cells to the loss of HMCES was recently reported and mechanistically defined to greater depth by Mehta et al, 2020 (PMC7313144). While it will be useful to the field that two groups independently arrived at the conclusion that A3A-expressing cells require HMCES for optimal fitness, it is difficult to be enthusiastic about this manuscript in the absence of additional insights, mechanistic or translational. 

Other specific comments:

1) To further support a role for targeting HMCES in A3A-overexpressing cells, it would be interesting to assess whether the catalytic activity of HMCES is involved in promoting the fitness of A3A-expressing cells.

2) Similarly, the authors should demonstrate experimentally that silencing A3A in a cancer cells displaying a APOBEC mutation signature reduces dependency on HMCES.

3) The authors state that 'HMCES depletion results in synthetic lethality with A3A expression specifically in a TP53-mutant background'. This claim relies solely in the interpretation of their screening data in A549 cells and lacks experimental validation. As this one of the few novel aspects of this manuscript, it should be validated experimentally.

4) In a similar vein, what is the impact of the p53 status on the analysis of dependencies across the 76 lung and head-and-neck cancer cell lines with respect to HMCES essentiality? 

5) We strongly encourage the authors to provide, at minimum, the gene-level depletion scores for their screens so they can be independently analyzed.

6) The use of time-dependent depletion is an interesting approach to select for true-positives. However, I wonder if it improves the detection of "core" essential genes, which could be used to validate this approach. One worry I have is that time-dependency may favor the identification of sgRNAs that target genes that code for unstable proteins or messenger mRNAs as those may be the first to show depletion at the earliest time point.

Minor issues

- Fig. 2B lacks a label on the X-axis

- On p4, the authors write MCM8/9-HROB/MCM8IP: this is confusing as it seems to suggest that HROB and MCM8IP are different proteins 

Reviewer #2:

Biayna and coworkers perform a very interesting set of CRISPR screens to identify genetic dependencies with overexpression of the single-stranded DNA deaminase APOBEC3A (A3A). A strength of the study is screening in multiple lines as well as in the presence/absence of p53, which has been shown previously to influence CRISPR results as well as A3A induced genotoxicity. Results with artificial overexpression are very clear. However, a major weakness limiting overall impact is not showing that cancer cell lines with endogenous A3A levels have evolved to be dependent on HMCES function. As it stands, artificially high A3A levels may cause genotoxic stresses that are not normally occurring/accumulating in tumor cells.

1) This manuscript shows that A3A overexpression (by Dox system) leads to a synthetic dependency requiring HMCES expression. This genetic interaction may be artifactual and a critical missing experiment is demonstrating that endogenous A3A levels (or "naturally" upregulated levels in cancer lines) are also synthetic with endogenous HMCES expression. In other words, is the viability of cancer cell lines with endogenous A3A expression similarly dependent upon HMCES?

2) Both A3A and A3B have been implicated in cancer mutagenesis. This begs the question as to whether A3B overexpression also has a synthetic lethal interaction with HMCES knockout/down? Addressing this may also help address point #1 above, as A3B is overexpressed in far more tumor cell lines than A3A.

3) The authors cite a recent Cell Reports paper that was the first to demonstrate a synthetic lethal interaction between A3A and HMCES (ref 47). This paper should be a focal point in the discussion (similarities/differences, etc). I don't think this prior work ruins the overall impact of the present study as the extensive CRISPR screening data are valuable.

4) Data accessibility - perhaps I missed it, but the supplement should include results for all genes covered in each CRISPR screen. The long term impact of this paper may be the large data sets that can be compared, contrasted, and followed-up by the broader community. 

Reviewer #3:

This paper describes the genetic connection between HMCES and the overexpression of APOBEC in lung cancer cells. Indeed, the authors identify HMCES as required to maintain viability in cancer cell with high levels of APOBEC. This synthetic lethality is observed directly in several lung cancer cell lines, but is validated further using available cancer expression data. Finally, the authors observed a synergistic effect of the combination of HMCES depletion with several DNA damaging agents and DNA repair factor inhibitors currently in use or in clinical trials for cancer treatment.

In general, the data presented in the paper are solid, the experiments are the appropriate, they are well executed and fully support the authors claim. Thus, scientifically I have no doubts of the manuscript validity. On the other hand, the main message of the paper is a compelling genetic interaction between APOBEC and HMCES. Although there is little insight on the molecular mechanisms behind this interaction, further than the known fact that HMCES protect abasic sites, it is clear that these observations might have relevance in cancer treatment. For this reviewer is hard to judge if the potential oncological treatment value is enough to overcome the lack of mechanistical information. Thus, what I can say is that scientifically I think the data support the main claims of the authors, but the editor should consider if the advance is enough to grant the publication in PLOS Biology

Other points:

1. HMCES is, among the original candidates, the one that show a clear differential behavior regarding TP53 status. However, in the studies using available cancer data there is no mention to this protein status. Could the authors separate the data between TP53 positive and negatives and see if this dependence on this protein is validated?

---

## [Decision Letter · Decision Letter 2]

17 Feb 2021

Dear Dr Stracker,

Thank you for submitting your revised Research Article entitled "Loss of HMCES is synthetic lethal with APOBEC activity in cancer cells" for publication in PLOS Biology. I have now obtained advice from two of the original reviewers and have discussed their comments with the Academic Editor. 

Based on the reviews, we will probably accept this manuscript for publication, provided you satisfactorily address the remaining points raised by the reviewers. Please also make sure to address the following data and other policy-related requests.

IMPORTANT:

a) Please attend to the remaining requests from reviewer #2.

b) Please could you make the title a bit more explicit? We suggest "Loss of HMCES is synthetic lethal with cytosine deaminase APOBEC3A activity in cancer cells" - if you can think of a 2- or 3-word phrase that encapsulates what HMCES is/does (I couldn't) then it would also be helpful to include that in the title.

c) Please address my Data Policy requests (further down) by supplying the underlying data for the Figs and citing its location clearly in the legends.

We expect to receive your revised manuscript within two weeks. 

*Published Peer Review History*

*Early Version*

Sincerely,

Roli Roberts

Senior Editor,

rroberts@plos.org,

PLOS Biology

DATA POLICY:

Regardless of the method selected, please ensure that you provide the individual numerical values that underlie the summary data displayed in the following figure panels as they are essential for readers to assess your analysis and to reproduce it: Figs 1AC, 2BCD, 3AB, 4ABCDEFGHI, 5ACD, 6B, S2, S3ABCD, S4B, S5AB, S6ABC, S7, S8, S9, S10ABC, S11C. I note that some of these data might be in your 13 Supplementary Tables; if so, please clarify. NOTE: the numerical data provided should include all replicates AND the way in which the plotted mean and errors were derived (it should not present only the mean/average values).

REVIEWERS' COMMENTS:

Reviewer #1:

First I need to mention that I was not aware that the manuscript was under "scoop protection" during the initial submission, which explained some of my initial comments. That being said, the authors did a very good job addressing my comments and I am happy to recommend publication.

Reviewer #2:

In this work by Byana and colleagues performed a CRISPR screen to identify synthetic lethal combinations with high levels of APOBEC3A expression in cell lines derived from lung adenocarcinoma (A549 and LXF-289). They identified HMCES as permissive factor for A3A mutagenesis, since depletion of HMCES leads to A3A-driven cell death as a result of hypermutations and increased levels of DNA damage. Moreover, they demonstrate that this phenotype is dependent on the TP53 status of the cells. In fact, cells mutated or lacking functional p53 are more sensitive to the HMCES depletion to A3A mediated damage. 

The majority of weak points of this manuscript were addressed in the first round of revisions and the authors answered to the majority of questions. Even though the synthetic lethality of A3A activity and HMCES depletion was previously described by the Cortez lab (Mehta et al, 2020, Cell Reports), the authors have notably improved the quality and novelty of the manuscript by performing a secondary CRISPR screen in HMCES KO cell lines. This screen provided insights into other genetic interactions of HMCES in the context of A3A activity and confirmed that the sensitivity of the depletion of HMCES is dependent on A3A activity. In fact, they show that A3A is among the top hits in this screen. Furthermore, they demonstrated that cancer cells are more sensitive to HMCES depletion when A3A expression is high and cells are lacking TP53.

Overall, the manuscript is well-revised and a strong contribution to the literature. My final (minor) point is that the discussion should be a bit better balanced to address why previously described A3A synthetic lethal combinations do not appear in the present work. For instance, multiple labs have reported a synthetic lethal interaction with A3A and ATR inhibition but, curiously, ATR does not appear to come up in any of the screens here (though it is used to help functionally validate the present work). These differences should be included in the discussion of the present manuscript.

---

## [Editor Report · Decision Letter 3]

8 Mar 2021

Dear Dr Stracker,

On behalf of my colleagues and the Academic Editor, Tanya Paull, I'm pleased to say that we can in principle offer to publish your Research Article "Loss of the abasic site sensor HMCES is synthetic lethal with activity of the APOBEC3A cytosine deaminase in cancer cells" in PLOS Biology, provided you address any remaining formatting and reporting issues. These will be detailed in an email that will follow this letter and that you will usually receive within 2-3 business days, during which time no action is required from you. Please note that we will not be able to formally accept your manuscript and schedule it for publication until you have made the required changes.

PRESS: We frequently collaborate with press offices. If your institution or institutions have a press office, please notify them about your upcoming paper at this point, to enable them to help maximise its impact. If the press office is planning to promote your findings, we would be grateful if they could coordinate with biologypress@plos.org. If you have not yet opted out of the early version process, we ask that you notify us immediately of any press plans so that we may do so on your behalf.

Thank you again for supporting Open Access publishing. We look forward to publishing your paper in PLOS Biology. 

Sincerely,

Roli Roberts

Roland G Roberts, PhD 

Senior Editor 

PLOS Biology